# Mitochondrial damage triggers the concerted degradation of negative regulators of neuronal autophagy

Bishal Basak [1,2] & Erika L. F. Holzbaur [1,2] ✉

Mutations that disrupt the clearance of damaged mitochondria via mitophagy are causative for neurological disorders including Parkinson's. Here, we identify a Mitophagic Stress Response (MitoSR) activated by mitochondrial damage in neurons and operating in parallel to canonical Pink1/Parkin-dependent mitophagy. Increasing levels of mitochondrial stress trigger a graded response that induces the concerted degradation of negative regulators of autophagy including Myotubularin-related phosphatase (MTMR)5, MTMR2 and Rubicon via the ubiquitin-proteasome pathway and selective proteolysis. MTMR5/MTMR2 inhibit autophagosome biogenesis; consistent with this, mitochondrial engulfment by autophagosomes is enhanced upon MTMR2 depletion. Rubicon inhibits lysosomal function, blocking later steps of neuronal autophagy; Rubicon depletion relieves this inhibition. Targeted depletion of both MTMR2 and Rubicon is sufficient to enhance mitophagy, promoting autophagosome biogenesis and facilitating mitophagosome-lysosome fusion. Together, these findings suggest that therapeutic activation of MitoSR to induce the selective degradation of negative regulators of autophagy may enhance mitochondrial quality control in stressed neurons.

Mitochondria are the primary organelles responsible for fueling the cell's metabolic needs by producing adenosine triphosphate (ATP)[1,2]. Neurons, by virtue of their polarized nature and high metabolic activity, are particularly dependent on mitochondria to supply energy to regulate neurotransmission, calcium homeostasis, and neural plasticity[3–5]. This elevated metabolic activity makes neuronal mitochondria more susceptible to damage; as neurons are post-mitotic, this damage can continue to accumulate over the life span of an organism. The build-up of dysfunctional mitochondrial fragments is toxic for neurons, as it leads to the generation of reactive oxygen species (ROS), pro-inflammatory signaling molecules, and may trigger cell death, all of which lead to neurodegeneration[6].

Neurons employ several quality control mechanisms to prevent the accumulation of damaged mitochondria and hence preserve mitochondrial physiology[7,8]. One of the critical processes that regulates organelle turnover and homeostasis in eukaryotic cells is autophagy. Under basal conditions, neuronal autophagy initiates primarily at synaptic sites and at the axon terminal, where a growing phagophore engulfs nearby cytoplasmic components[9,10]. The autophagosome matures as it traffics retrogradely to the soma where its fuses with the acidic lysosome to mediate degradation of its cargo[9,11].

Under basal conditions, mitochondria constitute ~10–20% of the cargos degraded by non-selective autophagy in the brain[12]. However, when mitochondria undergo depolarization or oxidative stress, their targeted removal is primarily driven by receptor-mediated selective autophagy, known as mitophagy[13–16]. In neurons, Pink1/Parkin-mediated mitophagy remains the most well-characterized pathway for mitochondrial quality control by autophagy[17]. In this pathway, mitochondrial damage induces the accumulation of PINK1 kinase on the outer mitochondrial membrane[18,19], which leads to the recruitment and activation of the E3 ligase Parkin[19–23]. Activated Parkin then ubiquitinates a plethora of mitochondrial proteins[24–30]. These ubiquitinated

[1]Department of Physiology, Perelman School of Medicine, University of Pennsylvania, Philadelphia, PA, USA. [2]Aligning Science Across Parkinson's (ASAP) Collaborative Research Network, Chevy Chase, MD, USA. ✉e-mail: holzbaur@pennmedicine.upenn.edu

proteins serve as a binding platform for recruitment of autophagy receptors[31–39] to promote the formation of an engulfing autophagosome that sequesters the damaged mitochondria away from the cytosol. Mutations in genes involved in this pathway lead to Parkinson's disease (PD)[40,41], Amyotrophic lateral sclerosis (ALS)[42,43] and Frontotemporal dementia (FTD)[44,45]. These observations suggest that mitophagy is a critically important pathway to maintain neuronal health; however, there is a limited understanding of how this pathway may be regulated in neurons.

Accumulating data suggest that the regulatory pathways that control autophagy in neurons are cell type-specific, as they are relatively resistant to autophagy inducers such as mTOR inhibition or amino acid starvation[46,47]. This resistance is partly mediated by the expression of Myotubularin-related phosphatase 5 (MTMR5), a protein that negatively regulates autophagy in neurons[48]. MTMR5 belongs to the myotubularin family of proteins, which function as phosphatidylinositol 3-phosphate (PI3P) and phosphatidylinositol 3,5-phosphate [PI(3,5)P$_2$] phosphatases. Proteins in this family, such as MTMR6, MTMR8, MTMR9, MTMR14 have been reported to repress autophagy in different cell types[49–51]. MTMR5, a catalytically inactive member of the family, is particularly enriched in neurons[48] and has been shown to modulate the localization and enzymatic activity of the active MTMR2 phosphatase[52]. The MTMR5-MTMR2 complex collectively represses early steps of autophagosome biogenesis by dephosphorylating PI3P produced by the action of VPS34[48]. Consistent with this model, knockdown of MTMR5 or MTMR2 in iPSC-derived neurons results in an increase in autophagosome number during mTOR inhibition[48].

Another negative regulator known to down-regulate autophagy in non-neuronal cells is Rubicon[53,54]. Rubicon was first identified as a subunit of the class III phosphatidylinositol 3-kinase complex II (PI3KC3-C2). PI3KC3-C2 includes the subunits UVRAG, VPS34, VPS15 and BECN1, and plays critical roles in phagophore expansion and endosome maturation[55–57]. Rubicon associates with a subpopulation of PI3KC3-C2 to block recruitment of the complex to the growing membrane[58]. Consistent with this, overexpression of Rubicon in non-neuronal cells reduces PI3P formation[58] and autophagosome number[53,54,58]. Rubicon also binds to RAB7-containing late endosomes and lysosomes and has been implicated in regulating membrane trafficking and reducing autophagic flux[53,54,59,60]. A recent study in HeLa cells revealed that phosphorylation of RAB7 upon mitochondrial damage induces the exchange of Rubicon for another RAB7-binding protein, Pacer, which is a positive regulator of autophagy. This exchange allows for a modest increase in mitochondrial turnover by autophagy following mitochondrial depolarization, supporting a role for Rubicon in repressing organellar homeostasis[61].

In the nervous system, recent findings indicate that systemic deletion of Rubicon in mice reduces the accumulation of phosphorylated α-synuclein upon injection of its native unphosphorylated form in the brain striatum[62], a primary pathological hallmark of PD. Rubicon levels are also reported to be higher in the lumbar spinal cord region of ALS patients[63]. These observations thereby necessitate a detailed molecular understanding of the role of Rubicon in regulating neuronal homeostasis by autophagy.

Here, we demonstrate that in response to mitochondrial damage in neurons, MTMR5, MTMR2 and Rubicon undergo concerted degradation in a graded manner, independent of Pink1/Parkin activity. Upon mitochondrial stress, we find that Rubicon is ubiquitinated and targeted to the proteasome for degradation; MTMR2 and MTMR5 are also ubiquitinated in response to mitochondrial stress, but their degradation involves both the proteasome and calcium-dependent proteolysis. We term the selective degradation of these proteins in response to mitochondrial damage as the Mitophagic Stress Response (MitoSR), and hypothesize that this response accelerates mitophagic flux by degrading the negative regulators of autophagy. Consistent with recent findings that MTMR5/MTMR2 negatively regulate

autophagosome biogenesis[48], we find that loss of this myotubularin complex enhances mitochondrial engulfment in response to oxidative damage in neurons. In contrast, under basal conditions, Rubicon is recruited to neuronal lysosomes via RAB7, blocking lysosomal function and thereby inhibiting autophagosome maturation. We find that targeted depletion of these negative regulators significantly increases mitochondrial turnover in neurons under both basal conditions and upon induction of mild oxidative stress. Thus, we propose that therapeutic interventions directly targeting negative regulators of autophagy may promote clearance of damaged mitochondria in patients suffering from neurodegenerative diseases such as PD and ALS, where mitophagy is compromised.

## Results

### Mitochondrial stress initiates the degradation of negative regulators of autophagy

The process of autophagy is critical for neuronal survival, yet neurons are largely resistant to major changes in basal autophagy flux. Recent progress has identified several negative regulators of autophagy, including MTMR5, MTMR2[48] and Rubicon[53,54,58,61] that can reduce autophagic flux in different cellular systems. This led us to ask whether, during periods of cellular stress when there is a need to upregulate autophagy, can neurons bypass these autophagy suppressors? To test this idea, we treated mouse embryonic cortical neurons with increasing doses of Antimycin A (Ant A), a well-characterized mitochondrial depolarizing agent that inhibits complex III of the electron transport chain, generating localized ROS[33,36,64,65]. Levels of MTMR5, MTMR2 and Rubicon remain unaltered under vehicle (ethanol-EtOH) treated conditions or low doses (3 nM) of Ant A (Fig. 1A–D). However, with increasing Ant A concentrations (15 nM, 30 nM), we observed dramatic decreases in expression levels of MTMR5, MTMR2 and Rubicon. For MTMR2, we could also detect a parallel increase in the appearance of a degraded product that migrates just beneath the intact protein on western blots. For Rubicon, we observed a second, faster-migrating band on western blots that may represent a second isoform (Supplemental S1); as the expression of both of these bands decreased in parallel in response to mitochondrial damage, we focused on the more prominent band in our analysis. Together, these observations indicate that there is a concerted decrease in the levels of three negative regulators of autophagy, MTMR5, MTMR2 and Rubicon, in response to increasing concentrations of Ant A. These findings suggest that mitochondrial ROS production triggers the dose-dependent degradation of these proteins.

Pink1/Parkin-mediated mitophagy is a primary degradative mechanism to remove dysfunctional mitochondria from neurons[17]. Thus, we tested whether degradation of these negative regulators is dependent on Pink1/Parkin-mediated mitophagy. For this, we treated embryonic cortical neurons from *Parkin*[-/-] mice with increasing concentrations of Ant A. As observed for neurons from wild-type mouse embryos, we did not see degradation of the three negative regulators in *Parkin*[-/-] neurons treated with either vehicle (EtOH) or a low dose (3 nM) of Ant A. (Fig. 1E–H). Again, as seen for wild-type neurons, at higher Ant A concentrations (15 nM, 30 nM), we saw the concerted degradation of MTMR5/2 and Rubicon in *Parkin*[-/-] neurons. Collectively, these results indicate that the damage-induced degradation of these autophagy suppressors is independent of Parkin activity.

The two major degradative pathways in eukaryotic cells are autophagy and the proteasomal system. Given the association of these proteins with the negative regulation of autophagy, we first asked if blocking autophagy might rescue levels of MTMR5, MTMR2 and Rubicon in neurons undergoing mitochondrial stress. We treated neurons with Bafilomycin A1 (Baf A1) for 1 hour prior to adding either vehicle or Ant A for another 2 hrs in the presence of Baf A1. We found that the Baf A1-induced block in autophagosome-lysosome fusion did not rescue the loss of MTMR5/2 or Rubicon induced by mitochondrial

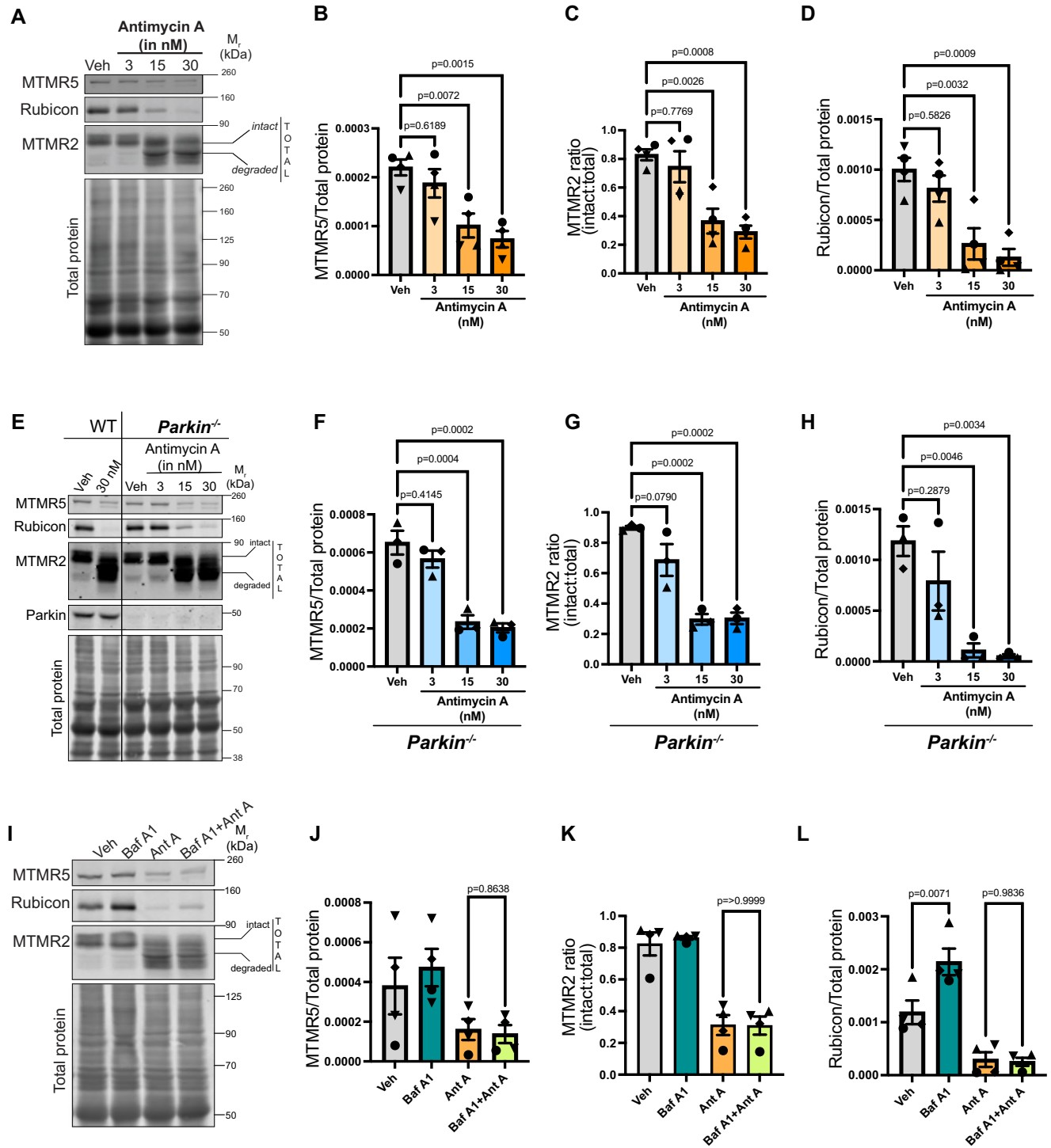

damage (Fig. 1I–L). Instead, we observed an approximately 2-fold increase in Rubicon levels, but no increase in the levels of MTMR2/5, when neurons were treated with Baf A1 alone as compared to vehicle (Fig. 1I, L). These data indicate that Rubicon undergoes turnover under basal conditions in neurons, potentially via lysosomal degradation or basal autophagy. However, neither of these mechanisms mediates the degradation of Rubicon induced upon mitochondrial stress.

### Degradative mechanisms targeting negative regulators of autophagy in response to mitochondrial stress

Next, we tested if the degradation of MTMR5, MTMR2, and Rubicon is mediated by the proteasome. We treated neurons with MG132, a drug

that blocks proteasome activity, for 1 hr and then added either vehicle or Ant A along with MG132 for an additional 2 hrs. We found that inhibiting proteasome activity completely rescued the degradation of MTMR5, MTMR2 and Rubicon that is observed in control cells upon mitochondrial stress (Fig. 2A–D). We also saw an ~2-fold increase in Rubicon levels when neurons were treated with MG132 only, in the absence of Ant A, as compared to vehicle-treated neurons (Fig. 2A, D), suggesting that Rubicon turnover is tightly regulated in neurons by both autophagy (Fig. 1I, L) and the proteasomal pathway (Fig. 2A, D). In contrast, when we treated HeLa cells with either Baf A1 or MG132, we saw, respectively, either no increase (Supplemental S2A, B) or only a very minor change (Supplemental S2C, D) in Rubicon levels. These

**Fig. 1 | Mitochondrial stress initiates the concerted degradation of MTMR5, MTMR2 and Rubicon, independent of Parkin activity or autophagic flux.**
**A–D** MTMR5, MTMR2, and Rubicon are degraded in response to increasing concentrations of Ant A. **A** Representative western blots from lysates of wild type (WT) murine embryonic cortical neurons treated with vehicle (EtOH), or with increasing concentrations of Ant A (3 nM, 15 nM or 30 nM) for 2 hrs. **B** MTMR5 band intensity normalized to total protein from WT neurons treated with EtOH or increasing concentrations of Ant A. **C** Ratio of intact MTMR2 band intensity normalized to total (intact + degraded) band intensities from WT neurons treated with EtOH or increasing concentrations of Ant A. **D** Rubicon band intensity normalized to total protein from WT neurons treated with EtOH or increasing concentrations of Ant A (**A–D**: N = 4 experiments, One way ANOVA with Dunnett's multiple comparison test). **E–H** Concerted degradation of MTMR5, MTMR2, and Rubicon is also observed in cortical neurons from *Parkin*[−/−] mouse embryos. **E** Representative western blots from lysates of *Parkin*[−/−] murine embryonic cortical neurons treated with vehicle (EtOH), or with increasing concentrations of Ant A (3 nM, 15 nM or 30 nM) for 2 hrs. **F** MTMR5 band intensity normalized to total protein from *Parkin*[−/−] neurons treated with EtOH or increasing concentrations of Ant A. **G** Ratio of intact

MTMR2 band intensity normalized to total (intact + degraded) band intensities from *Parkin*[−/−] neurons treated with EtOH or increasing concentrations of Ant A. **H** Rubicon band intensity normalized to total protein from *Parkin*[−/−] neurons treated with EtOH or increasing concentrations of Ant A (**E–H**: N = 3 experiments, One way ANOVA with Dunnett's multiple comparison test). **I–L** Inhibition of autophagy does not block the degradation of MTMR5, MTMR2, or Rubicon in response to mitochondrial damage. **I** Representative western blots from lysates of WT cortical neuronal lysates treated with vehicle (DMSO) or 500 nM Bafilomycin A1 (Baf A1) for 1 hr followed by an additional treatment with vehicle (EtOH) or 15 nM Ant A for 2 hrs. **J** MTMR5 band intensity normalized to total protein from WT neurons treated with DMSO/Baf A1 and EtOH/Ant A. **K** Ratio of intact MTMR2 normalized to total (intact + degraded) band intensities from WT neurons treated with DMSO/Baf A1 and EtOH/Ant A. **L** Rubicon band intensity normalized to total protein from WT neurons treated with DMSO/Baf A1 and EtOH/Ant A (**I–L**: N = 4 experiments, **J, L**: one way ANOVA with Sidak's multiple comparison test, **K**: Kruskal–Wallis test). Error bars indicate SEM. Source data are provided as a Source Data file.

results indicate that Rubicon levels are tightly regulated in a cell-type-specific manner in neurons.

Proteins that are targeted to the proteasomal machinery for degradation are first conjugated to ubiquitin (Ub) or Ub-like proteins. Since we see accumulation of Rubicon in response to MG132 treatment in neurons, we tested if Rubicon is ubiquitinated in neurons, and whether this ubiquitination is increased in response to mitochondrial damage. For this, we treated neurons with vehicle (EtOH) or Ant A for 2 hrs, and immunoprecipitated Rubicon to enrich for ubiquitinated forms of the protein. Indeed, we were able to see higher molecular weight bands emerge that were positive for both Rubicon and Ub on western blots (Fig. 2E). More importantly, we were able to see a distinct band of ubiquitinated Rubicon emerge in response to mitochondrial damage (Fig. 2E). Our result shows a significant increase in the ratio of ubiquitinated to total Rubicon upon mitochondrial damage, indicating that this damage accelerates ubiquitination of Rubicon and its degradation by the proteasomal machinery (Fig. 2E, F). In parallel experiments, we immunoprecipitated MTMR2; this enrichment led to the appearance of higher molecular weight, ubiquitinated bands positive for MTMR2, although we did not see a significant increase in ubiquitination in response to mitochondrial stress (Supplemental S3A, B). This observation, in parallel with the apparent proteolytic cleavage of MTMR2 induced by oxidative stress (Fig. 1A), suggests that additional mechanisms may contribute to the degradation of MTMR2. We were unable to perform a parallel experiment for MTMR5 as the available antibody proved to be ineffective for immunoprecipitation.

To further substantiate our findings, we performed the complementary experiment, enriching for ubiquitinated proteins and then probing for the presence of MTMR5/2 and Rubicon in the immunoprecipitated fraction. For this experiment, we treated neurons with 10 μM MG132 for 1 hr to block proteasomal degradation of ubiquitinated proteins and then added Ant A for an additional 2 hrs in the presence of MG132 to trigger mitochondrial damage. Post-treatment, we pulled down Ub-bound proteins from the treated neuronal lysates using an Ub enrichment kit (see Methods). We saw significant enrichment of ubiquitinated proteins in the pull-down fraction when compared to pulldowns with control beads (Fig. 2G, H). As a positive control for this approach, we probed for Mitofusin-2 (MFN-2), as this mitochondrial outer membrane GTPase is ubiquitinated by Parkin as one of the initial steps following induction of PINK1/Parkin-dependent mitophagy[28,66]. Consistent with this, we detected enrichment of MFN-2 in the Ub-pull-down fraction, confirming the onset of mitophagy (Fig. 2G). Western blot analysis indicated that all three negative regulators of autophagy, MTMR5/2 and Rubicon (Fig. 2G, I–K) were enriched in the Ub-pull-down fraction.

Protein ubiquitination involves a cascade of three sequential enzymatic reactions that include a Ub-activating enzyme (E1), a Ub-conjugating enzyme (E2) and a Ub-ligase (E3)[67]. We asked if blocking the enzymatic conjugation of Ub to MTMR5, MTMR2, and Rubicon would prevent their degradation. For this, we treated neurons with PYR41, a drug that irreversibly blocks the activity of all Ub-E1 enzymes, for 1 hr and then added vehicle or Ant A along with PYR41 for an additional 2 hrs. We observed a complete rescue of Rubicon levels upon blocking E1 activity during mitochondrial stress (Supplemental S3C, D). For MTMR2, we observed a significant but partial rescue of its intact form when treated with PYR41 upon mitochondrial damage (Supplemental S3C, E) and a similar trend was observed with MTMR5 (Supplemental S3C, F).

Together, these observations suggest that Rubicon is primarily degraded by the proteasomal pathway in response to mitochondrial stress, but additional degradative mechanisms are likely to contribute to the loss of MTMR5 and MTMR2. Previous studies have brought into focus the role of calcium ($Ca^{2+}$) in regulating mitophagy[68–70]. To investigate whether the degradation of MTMR5/2 is $Ca^{2+}$ dependent, we treated neurons with two $Ca^{2+}$ chelating agents-EGTA and BAPTA. We observed complete rescue of the degradation of MTMR5 and MTMR2 upon treatment of neurons with EGTA during mitochondrial stress (Supplemental S3G–I). Based on this finding, we hypothesize that changes in cytosolic $Ca^{2+}$ levels induced by mitochondrial damage may activate $Ca^{2+}$-activated proteases that degrade MTMR2 and MTMR5. Further work will be required to identify these additional mechanisms that contribute to the degradation of MTMR5/2.

Rubicon, in contrast, is primarily cleared by the ubiquitin-proteasome system. Since this degradation is independent of the E3 ligase Parkin (Fig. 1E, H), we wondered what other E3 ligase might induce ubiquitination of Rubicon under basal conditions and in response to oxidative damage. Recent work in chondrocytes has shown that the E3 ligase HECTD1 regulates Rubicon levels via ubiquitination[71]. We asked whether HECTD1 regulates Rubicon levels in neurons by transducing primary cortical neurons at day in vitro (DIV) 1 with lentivirus encoding an shRNA against HECTD1 or a non-targeting control. We were able to deplete HECTD1 by more than 60% by DIV 6-8, and noted a 47% increase in Rubicon, indicating that HECTD1 regulates Rubicon levels in neurons under basal conditions (Supplemental S4A–C). To investigate if HECTD1 is essential for regulating Rubicon levels in response to mitochondrial damage, we depleted HECTD1 with lentiviral shRNA and then treated neurons with Ant A for 2 hrs. Under these conditions, we saw a small but not significant increase in the depleted levels of Rubicon relative to control neurons also treated with Ant A (Supplemental S4D–F). Thus, while HECTD1 regulates cellular levels of Rubicon under basal conditions, additional E3 ligases may either

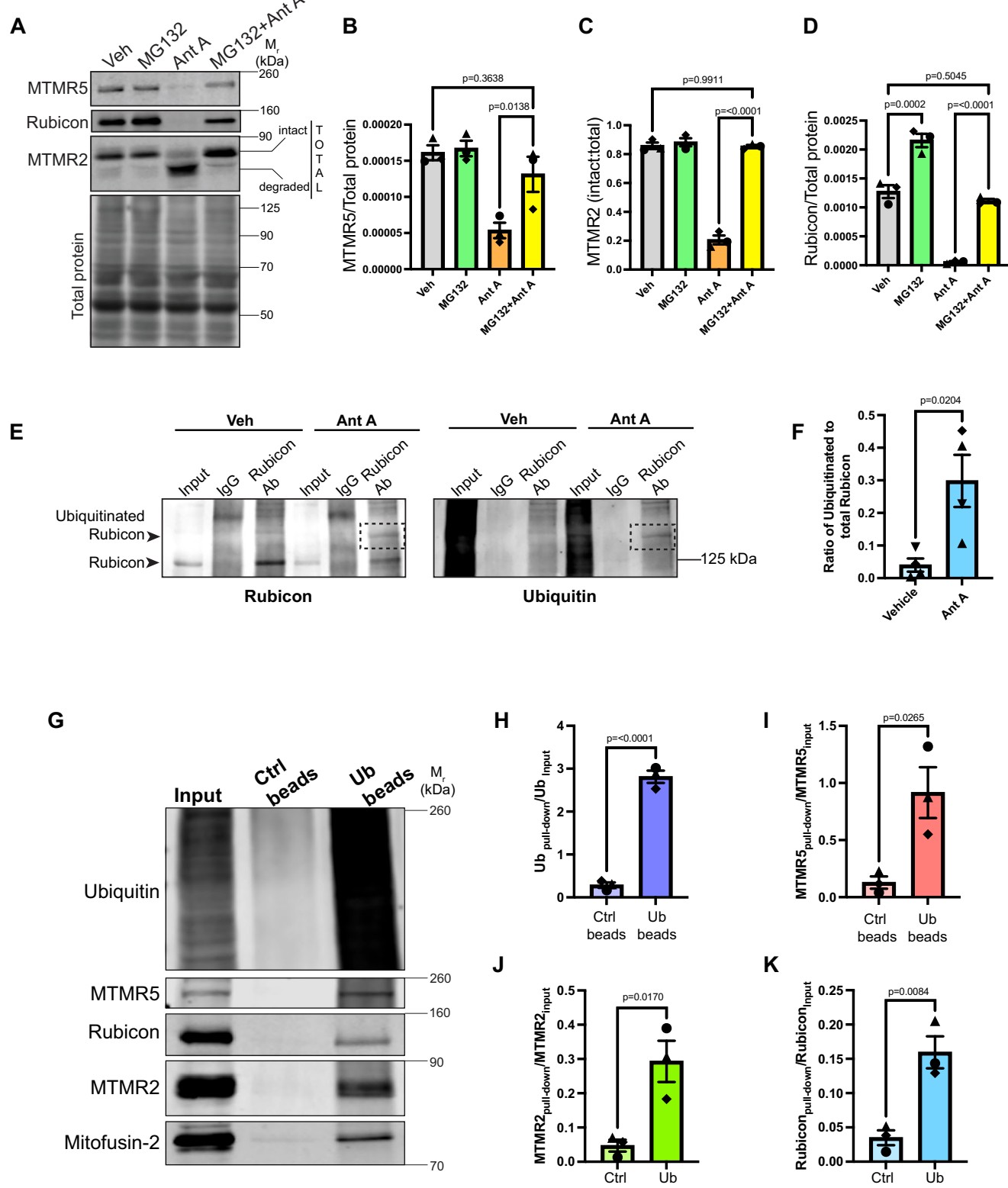

contribute to this regulation or compensate for the loss of HECTD1 function in response to mitochondrial damage. HECTD1 is a member of the 28-member HECT family of E3 ligases, so it is possible another family member may exhibit redundant substrate specificity. Indeed, the concept of multiple E3 ligases regulating protein levels during mitochondrial stress has been investigated before. In the absence of Parkin, the E3 ligase MUL1 is reported to regulate levels of the outer mitochondrial membrane protein MFN-2 to facilitate mitophagy[72]. Similarly, HUWE1, also a member of the HECT family of proteins, is known to ubiquitinate MFN-2 upon mitochondrial damage to facilitate mitophagy[73]. Further work testing candidate E3 ligases combinatorially is required to fully address this question.

**Fig. 2 | MTMR5, MTMR2 and Rubicon are ubiquitinated and degraded by the proteasome upon acute mitochondrial damage. A–D** Proteasomal inhibition blocks the concerted degradation of MTMR5, MTMR2, and Rubicon. **A** Representative western blots from lysates of WT murine embryonic cortical neurons treated with vehicle (EtOH) or 10 µM MG132 for 1 hr followed by an additional treatment with vehicle (EtOH) or 15 nM Ant A for 2 hrs. **B** MTMR5 band intensity normalized to total protein from WT neurons treated with EtOH/MG132 and EtOH/Ant A. **C** Ratio of intact MTMR2 normalized to total (intact + degraded) band intensities from WT neurons treated with EtOH/MG132 and EtOH/Ant A. **D** Rubicon band intensity normalized to total protein from WT neurons treated with EtOH/MG132 and EtOH/Ant A (**A–D**: $N = 3$ experiments, one way ANOVA with Sidak's multiple comparison test). **E, F** Mitochondrial damage accelerates ubiquitination of Rubicon in neurons. **E** Representative western blot from wild type cortical neuronal lysates treated with vehicle (EtOH) or 30 nM Ant A for 2 hrs and immunoprecipitated with Rubicon antibody or control Rabbit IgG. **F** Quantification

of the ratio of ubiquitinated to total Rubicon upon immunoprecipitation of Rubicon in neurons treated with EtOH or Ant A ($N = 4$ experiments, two-tailed unpaired $t$ test). **G–K** Negative regulators of autophagy are ubiquitinated in response to mitochondrial stress. **G** Representative western blot of ubiquitin enrichment assay from lysates of WT murine embryonic cortical neurons. Neurons were treated with 10 µM MG132 for 1 hr followed by an additional treatment with 15 nM Ant A for another 2 hrs (Ub beads = Ubiquitination affinity beads). **H** Ub enrichment following pull-down with Ub beads or control beads, normalized to intensity in the input lane. **I** MTMR5 enrichment following pull-down with Ub beads or control beads, normalized to intensity in the input lane. **J** MTMR2 enrichment following pull-down with Ub beads or control beads, normalized to its intensity in the input lane. **K** Rubicon enrichment after pull-down with Ub beads or control beads, normalized to its intensity in the input lane (**G–K**: $N = 3$ experiments, two-tailed unpaired $t$ test). Error bars indicate SEM. Source data are provided as a Source Data file.

## The degradation of MTMR5/2 and Rubicon represents a specific response to mitochondrial damage in neurons

We next asked whether the targeted degradation of negative regulators of mitophagy is a specific response or a part of more generalized loss of other mitophagy-associated proteins (Fig. 3A-H, and Supplemental S4G). For this, we first examined levels of the protein MFN-2, which is known to be degraded upon activation of Pink1/Parkin. We saw the degradation of MFN-2, confirming induction of mitochondrial damage (Fig. 3A, B). Next, we examined the effects of 30 nM Ant A treatment on Parkin (Fig. 3C), TBK1 (Fig. 3D), RAB7 (Fig. 3E), BNIP3 (Fig. 3F) and mitochondrial proteins, including COX2 (Fig. 3G) and Mitofilin (Fig. 3H), and found their levels to be generally unaffected.

We then looked at proteins that play an essential role in autophagosome biogenesis and maturation (Fig. 3I–M, and Supplemental S4H). We started by examining levels of VPS34, a component of PI3K complexes that promotes synthesis of PI3P on the growing autophagosome membrane, thus opposing MTMR5/2 function in autophagosome biogenesis[48,74,75]. We observed no significant change in cellular levels of VPS34 upon mitochondrial damage (Fig. 3 I). We also tested if levels of other key components of the autophagy machinery, including ATG5 (Fig. 3J), ATG7 (Fig. 3K), GABARAPs (Fig. 3L) and LC3 (Fig. 3M), were altered in response to mitochondrial damage, but found their total levels to remain unaffected upon treatment of cortical neurons with Ant A. We did note the appearance of an apparent cleavage product for ATG5, but total levels were found to be similar (Fig. 3J and Supplemental S4H–ATG5).

Next, we looked at lysosomal proteins (Fig. 3N–R, and Supplemental S4I), since this organelle plays a critical role in autophagosome maturation. We did see a significant reduction in the levels of the luminal lysosomal protease Cathepsin B (Fig. 3N, O), which may suggest enhanced lysosomal activity in response to mitochondrial damage, investigated in more detail below. However, we did not see significant changes in levels of other lysosomal proteins such as the subunit of the V-ATPase proton pump ATP6V1E1 (Fig. 3N, P) or LAMP1 (Fig. 3Q) and SCARB2 (Fig. 3R).

MTMR5 and MTMR2 belong to the 14-member family of myotubularin-related phosphatases, so we examined the effects of mitochondrial damage on the expression levels of two other MTMR family members previously implicated in autophagy and/or mitochondrial damage responses (Fig. 3S, T, and Supplemental S4J); MTM1 was implicated as responsive to changes in PINK1 activity in an unbiased proteomic screen[30] and MTMR14 (also known as JUMPY) is a previously characterized negative regulator of autophagy[49]. However, we did not see significant changes in the expression levels of either MTM1 or MTMR14 in primary cortical neurons treated for 2 hr with 30 nM Ant A (Fig. 3S, T), further confirming the specificity of the observed degradation of MTMR5 and MTMR2 in response to mitochondrial stress.

Finally, we compared these observations to levels of the housekeeping proteins (Fig. 3U–W, and Supplemental S4K) α/β-Tubulin, GAPDH, and Actin, which did not change in response to Ant A treatment of cortical neurons (Fig. 3U–W). Collectively, these results suggest that the degradation of MTMR5/2 and Rubicon upon mitochondrial damage represents the elimination of specific targets to release inhibition on mitophagy.

Next, we tested whether the degradation of these negative regulators is induced as a specific response to Ant A treatment or instead can be elicited by other mitochondrial damage-inducing paradigms. We treated primary cortical neurons with drugs that target different components of the mitochondrial electron transport chain: Oligomycin A (Oligo A) is an inhibitor of mitochondrial ATP synthase (Complex V) that blocks oxidative phosphorylation, while Carbonyl Cyanide m-Chlorophenylhydrazone (CCCP) is an uncoupler that interferes with the proton gradient across the inner mitochondrial membrane thereby disrupting mitochondrial membrane potential. Immunoblot analysis indicates that treatment of cortical neurons with Oligo A (500 nM, 10 µM) or CCCP (10 µM) was sufficient to induce the concerted degradation of MTMR5, MTMR2, and Rubicon (Fig. 4A–D). These findings confirm that elevating mitochondrial ROS production with Ant A, inhibiting ATP synthesis with Oligo A, or disrupting mitochondrial membrane potential with CCCP can trigger the activation of this stress response pathway.

We next asked if other forms of organellar damage induced a similar response in neurons. We targeted lysosomes by treating neurons for 2 hrs with 1 mM L-leucyl-L-leucine methyl ester (LLOMe), a well-characterized drug that disrupts the integrity of the lysosomal membrane, triggering lysophagy[76,77], which can be monitored by the recruitment of SQSTM1/p62 to damaged lysosomes[76]. Consistent with previous findings, we found that LLOMe treatment induced the formation of large p62 aggregates around lysosomes positive for LAMP1, thereby confirming induction of lysosomal stress (Supplemental S5A). However, under these conditions, we observed no changes in the expression levels of MTMR5/2 or Rubicon (Fig. 4E–H). In parallel experiments, we induced stress to the endoplasmic reticulum (ER) by treating neurons with 50 nM Tunicamycin for 3 hrs. Induction of ER stress was confirmed by elevated levels of the transcription factor ATF4[78] (Fig. 4I, J), but we saw no changes in the levels of MTMR5, MTMR2 or Rubicon under these conditions (Fig. 4I, K–M). Finally, we examined the effects of proteotoxic stress, which can be induced by treating neurons with 10 µM MG132 for 3 hrs to induce aggrephagy[79,80]. We observed the accumulation of ubiquitinated proteins (Supplemental S5B, C), thereby confirming proteotoxicity, but we did not see reduced levels of MTMR5, MTMR2 and Rubicon in neurons treated with MG132 (Supplemental S5D–G). Together, these results suggest that the degradation of MTMR5, MTMR2, and Rubicon is elicited in response to mitochondrial stress, but not to other cellular stressors, including those sufficient to induce lysophagy, ER stress, or

## Mitophagy and mitochondrial proteins

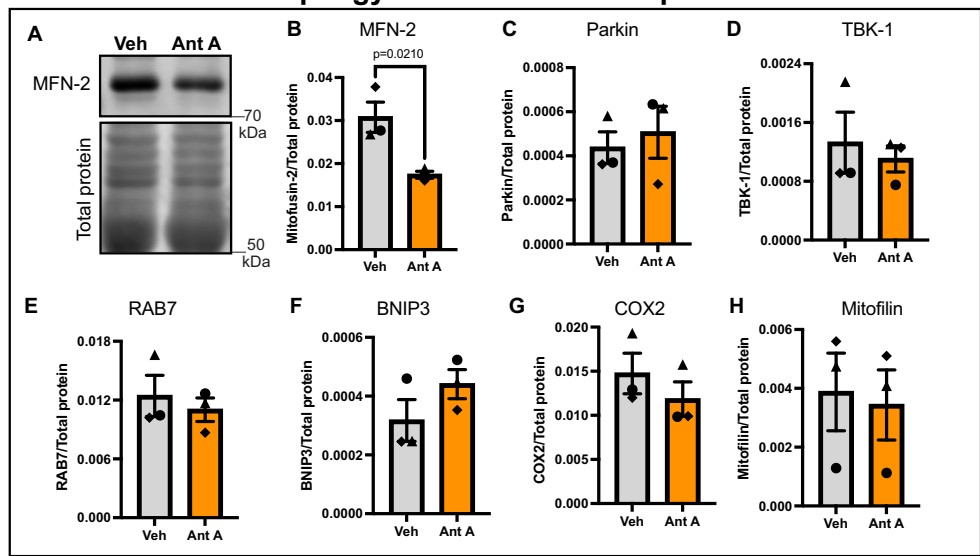

## Autophagy proteins

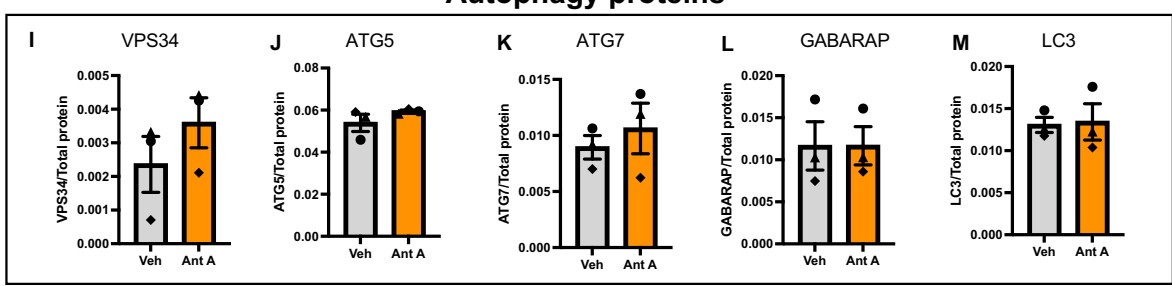

## Lysosomal proteins

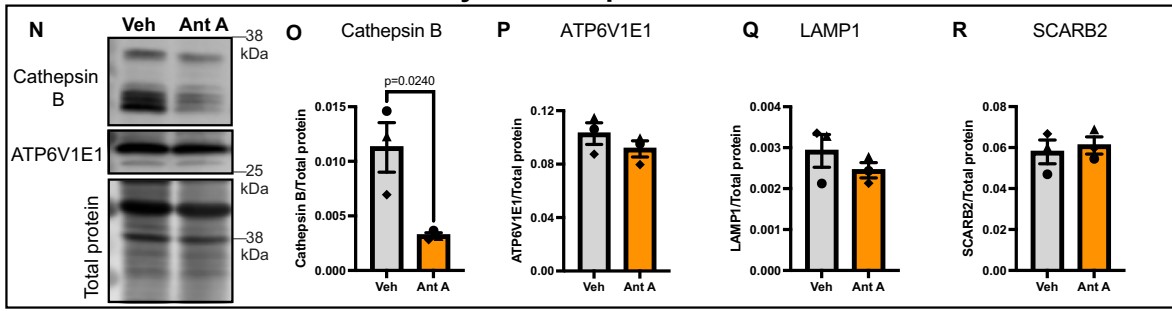

## MTMRs

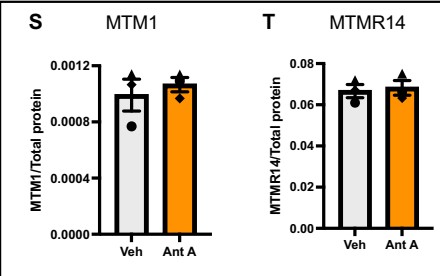

## House-keeping proteins

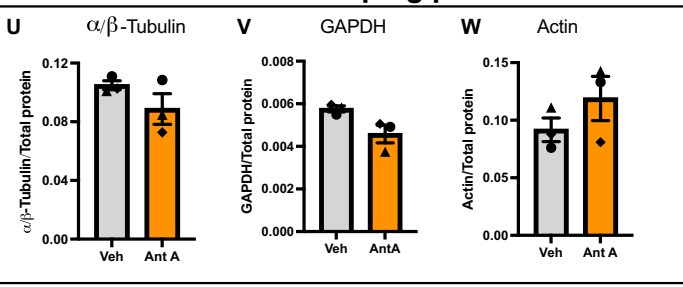

aggrephagy, supporting our hypothesis that this is a mitophagy-specific stress response.

Similar to neurons, astrocytes respond robustly to mitochondrial-damaging agents to trigger Pink1/Parkin-mediated mitophagy[81,82], although induction of this response typically requires higher drug concentrations than those found to be effective to induce mitophagy in neurons[36]. In line with this, when we treated primary murine astro-cytes with 10 μM Oligo A/5 μM Ant A, we detected a significant loss of MFN-2 as compared to vehicle (DMSO/Ethanol) treated cells (Fig. 4N, O), thereby confirming the onset of mitophagy. MTMR5 levels were very low in astrocytes under both conditions, consistent with the identification of MTMR5 as a neuron-enriched protein[48]. However,

**Fig. 3 | Mitochondrial damage does not impact the levels of majority of proteins congruent to mitophagy in neurons. A, B** Mitochondrial damage initiates degradation of MFN-2 as an outcome of Pink1/Parkin-mediated mitophagy. **A** Representative western blot of wild-type cortical neurons treated with vehicle (EtOH) or 30 nM Ant A for 2 hrs and probed for MFN-2. **B** Quantification of MFN-2 levels in neurons treated with vehicle (EtOH) or Ant A. **C–H** Mitochondrial damage does not influence the levels of additional mitophagy-associated proteins and mitochondrial proteins. Quantification of levels of other mitophagy proteins (**C** Parkin, **D** TBK1, **E** RAB7, **F** BNIP3) and mitochondrial proteins (**G** COX2, **H** Mitofilin) in neurons treated with vehicle (EtOH) or 30 nM Ant A for 2 hrs. **I–M** Levels of autophagy-promoting proteins are unaltered during mitochondrial damage. Quantification of levels of autophagy-associated proteins (**I** VPS34, **J** ATG5, **K** ATG7, **L** GABARAP, **M** LC3) in neurons treated with vehicle (EtOH) or 30 nM Ant A for 2 hrs. **N, O** Mitochondrial damage initiates degradation of Cathepsin B, presumably because of increased lysosomal function. **N** Representative western blot of wild type cortical neurons treated with vehicle (EtOH) or 30 nM Ant A for 2 hrs and probed for Cathepsin B and ATP6V1E1. **O** Quantification of Cathepsin B levels in neurons treated with vehicle (EtOH) or Ant A. **P–R** Lysosomal membrane proteins are unaffected during mitochondrial stress. Quantification of levels of lysosomal membrane proteins (**P** ATP6V1E1, **Q** LAMP1, **R** SCARB2) in neurons treated with vehicle (EtOH) or 30 nM Ant A for 2 hrs. **S, T** Mitochondrial damage does not target other proteins of the MTMR family. Quantification of levels of two members of the MTMR family (**S** MTM1, **T** MTMR14) in neurons treated with vehicle (EtOH) or 30 nM Ant A for 2 hrs. **U–W** House-keeping proteins are unaltered upon mitochondrial damage in neurons. Quantification of levels of house-keeping proteins (**U** α/β-Tubulin, **V** GAPDH, **W** Actin) in neurons treated with vehicle (EtOH) or 30 nM Ant A for 2 hrs. (All panels: for statistical analysis, two-tailed unpaired *t* tests were performed for all proteins except for COX2, BNIP-3 and TBK-1. Mann–Whitney test was performed for these three proteins because the data did not show a normal distribution. Error bars indicate SEM). Source data are provided as a Source Data file.

MTMR2 (Fig. 4N, P) and Rubicon (Fig. 4N, Q) levels were more robust and remained comparable across conditions, with no evidence of targeted degradation upon mitochondrial damage. Collectively, these findings indicate that mitochondrial damage in astrocytes triggers Pink/Parkin-mediated mitophagy without degrading MTMR5/2 and Rubicon. Likewise, in a well-characterized cell model used extensively to study mitophagy, HeLa cells transfected with exogenous Parkin[33], upon treatment with Ant A/Oligo A or vehicle (DMSO/Ethanol), we did not detect the degradation of the three negative regulators, although MFN-2 levels were reduced thus confirming induction of mitochondrial damage and initiation of Pink1/Parkin mitophagy (Supplemental S5H–L).

Taken together, these results indicate that while Pink1/Parkin mitophagy was successfully triggered by mitochondrial damage in all three cell types examined, the selective degradation of MTMR5/2 and Rubicon was only seen in neurons. Thus, we conclude that activation of this response is specific to mitochondrial stress in neurons and predominantly targets negative regulators of autophagy. Hence, we term this the MitoSR and propose that MitoSR is a neuron-specific cell-protective response to acute mitochondrial damage.

## Rubicon localizes to somal lysosomes and suppresses lysosomal activity in neurons

Induction of mitochondrial damage leads to the targeted degradation of MTMR5, MTMR2, and Rubicon, so we asked how the loss of these three negative regulators might enhance mitochondrial degradation. MTMR5 and MTMR2 are known to act together to repress the early steps of neuronal autophagy by inhibiting PI3P formation on autophagosomes[48], and thus degradation of these proteins is predicted to enhance autophagosome biogenesis. However, the role of Rubicon in regulating autophagy in neurons has not yet been investigated.

First, we probed for endogenous Rubicon and found an ~3.5-fold enrichment in neurons relative to HeLa cells (Fig. 5A, B). To examine the subcellular localization of Rubicon, we transfected neurons with an EGFP-tagged construct of human Rubicon and performed live cell imaging. We found Rubicon to form ring-like punctate structures that were predominantly distributed within the neuronal soma (Fig. 5C); punctate structures were also observed in both dendrites (Fig. 5D) and axons (Fig. 5E), but less frequently. We co-transfected neurons with an autophagosome marker (mCherry-LC3) and the lysosomal marker (LAMP1-Halo) along with EGFP-Rubicon. Almost 60% of EGFP-Rubicon puncta within the soma co-localized with LAMP1-Halo, while only 10% of EGFP-Rubicon puncta colocalized with mCherry-LC3 (Fig. 5F, G). Thus, in neurons, as well as in the non-neuronal lines A549 and HeLa cells[53,54], Rubicon is localized to lysosomes and late endosomes.

Rubicon binds to Rab7 via its C-terminal RH domain (Fig. 5H), and this interaction negatively impacts endocytic trafficking and autophagosome maturation in non-neuronal cells such as HEK293 or HeLa[59–61]. Within this RH domain, mutations of three conserved cysteine residues to glycine (*C912G*, *C915G*, *C923G*) and the histidine residue to leucine (*H920L*) (Rubicon[CGHL]) result in a complete loss of the Rubicon-Rab7 interaction[59,60]. To test if this domain determines Rubicon's localization in neurons, we introduced these mutations into full-length Rubicon and expressed this construct in neurons (Fig. 5H), co-expressing either mCherry-Rubicon[WT] or mCherry-Rubicon[CGHL] along with GFP-RAB7 and LAMP1-Halo. While Rubicon[WT] showed a strong membrane association and co-localized with both RAB7 and LAMP1, Rubicon[CGHL] remained cytosolic, no longer co-localizing with either RAB7 or LAMP1 (Fig. 5I–K). This observation suggests that the RH domain of Rubicon is critical for the association of Rubicon with lysosomes in neurons, mediating an interaction with RAB7 as demonstrated by structural studies[60].

Rubicon has been reported to block autophagosome maturation by inhibiting the recruitment of PI3KC3-C2 complex to the growing autophagosome membrane[58]. In neurons, however, we saw a striking association of Rubicon with lysosomes rather than autophagosomes (Fig. 5G). Thus, we asked if Rubicon influences lysosomal physiology. We knocked down endogenous Rubicon by nucleofection with a targeted shRNA plasmid, achieving ~60% depletion of the endogenous protein (Supplemental S6A, B). We labeled lysosomes using Lysotracker Green, which fluoresces in the acidic environment of the lysosome, and autophagosomes using the autophagy dye DAPRed, which incorporates into the autophagosome membrane and fluoresces in a hydrophobic environment[83–85]. Compared to control neurons, depletion of Rubicon resulted in an ~2.5-fold increase in the number of lysosomes marked by Lysotracker Green (Fig. 6A, B). We also saw a significant increase in the number of autophagosomes upon knockdown of Rubicon in neurons (Fig. 6A, C). It is important to note that the DAPRed dye is incorporated into the autophagosome membrane and will remain fluorescent following autophagosome-lysosome fusion; thus, the dye does not specifically distinguish between autophagosomes and autolysosomes, so that the increase in DAPRed puncta upon Rubicon knockdown could be reflective of either enhanced autophagosome biogenesis or more autolysosomes. Indeed, when we measured colocalization of DAPRed and Lysotracker puncta as a readout of autophagosome-lysosome fusion, we found an ~2.7-fold increase in the number of autolysosomes per soma when Rubicon was depleted from neurons (Fig. 6A, D).

Our data demonstrate that removal of Rubicon results in an increase in the number of lysosomes marked by Lysotracker. This could be indicative of more lysosomal biogenesis upon removal of Rubicon, and/or could suggest that Rubicon represses lysosomal acidification, so that Rubicon knockdown results in an increase in the fraction of acidified lysosomes. To determine if Rubicon influences lysosomal biogenesis, we knocked down Rubicon and probed for

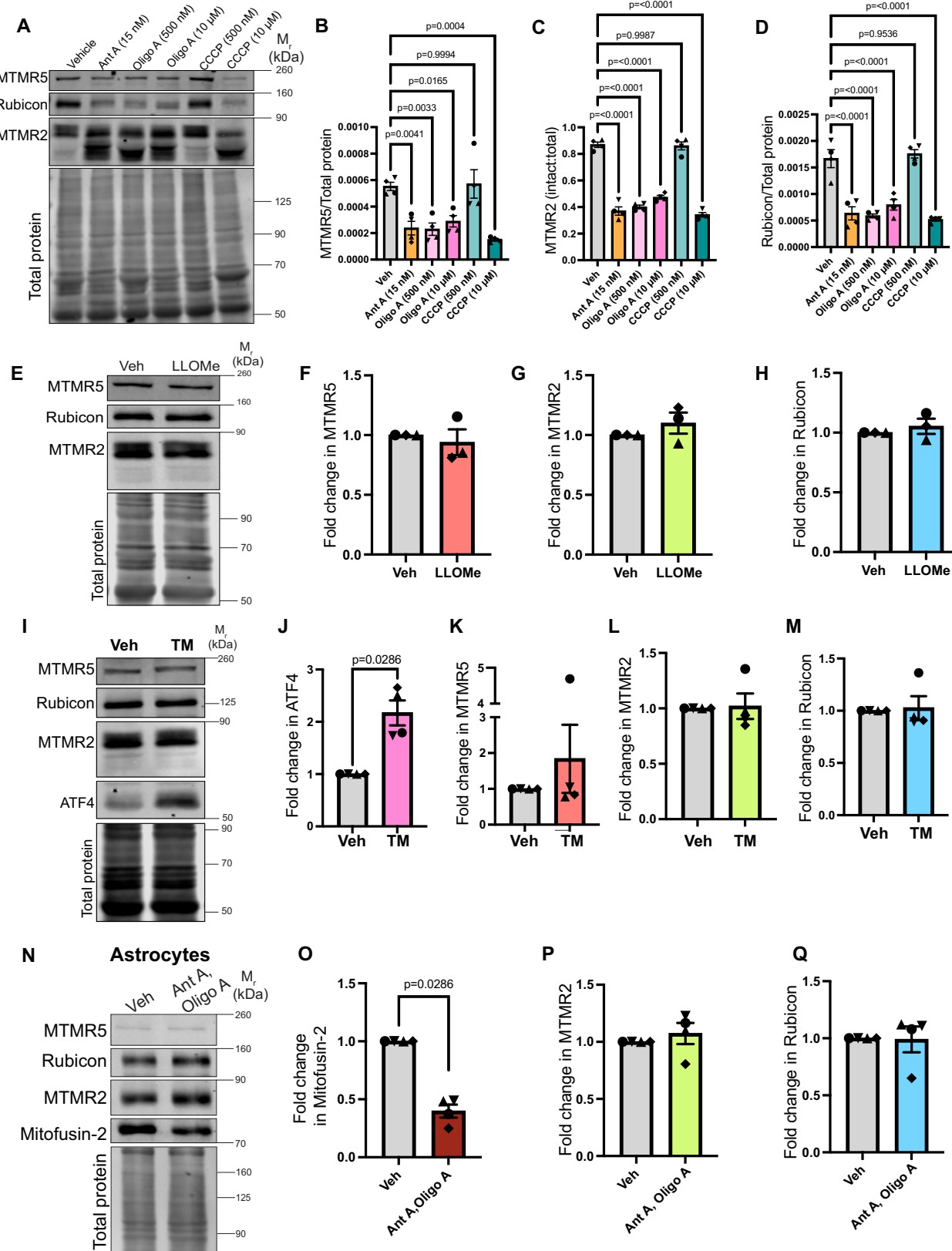

levels of the lysosomal proteins LAMP1 and SCARB2. Indeed, we observed a significant increase in the levels of both proteins upon depletion of Rubicon (Supplemental S6C–E), thereby suggesting that Rubicon negatively regulates lysosomal biogenesis. Next, we tested the impact of Rubicon on lysosomal function. For this experiment, we treated neurons with two lysosome labeling dyes, LysoPrime Green

and pHLys Red (see methods). While LysoPrime Green is pH resistant and thus labels the total pool of lysosomes, pHLys Red diffuses out of the organelle upon deacidification and thus preferentially labels the more acidic pool of lysosomes. We labeled neurons sequentially with these two dyes and then used 75 nM Baf A1 to partially deacidify lysosomes for 30 mins to increase the dynamic range of the assay

**Fig. 4 | Activation of the MitoSR is specific to mitochondrial damage in neurons and targets negative regulators of autophagy. A–D** Concerted degradation of MTMR5, MTMR2, and Rubicon is mediated by diverse triggers of mitophagy. **A** Representative western blot from WT cortical neuronal lysates treated with vehicle (EtOH, DMSO), Ant A, Oligo A, or CCCP for 2 hrs (concentration of drugs as indicated in the image). **B** MTMR5 levels normalized to total protein upon treatment of WT neurons with vehicle, Ant A, Oligo A, or CCCP for 2 hrs. **C** Ratio of intact MTMR2 normalized to total (intact + degraded) upon treatment of WT neurons with vehicle, Ant A, Oligo A, or CCCP for 2 hrs. **D** Rubicon band intensity normalized to total protein upon treatment of WT cortical neurons with vehicle, Ant A, Oligo A, or CCCP for 2 hrs (**A–D**: $N = 4$ experiments, one-way ANOVA with Dunnett's multiple comparison test). **E–H** Lysosomal damage does not trigger the degradation of the negative regulators of autophagy. **E** Representative western blot from lysates of WT murine embryonic cortical neurons treated with vehicle (EtOH) or 1 mM LLOMe for 2 hrs. **F** Fold change in MTMR5 levels upon treatment of WT neurons with LLOMe as compared to with vehicle. **G** Fold change in MTMR2 levels upon treatment of WT neurons with LLOMe as compared to with vehicle. **H** Fold change in Rubicon levels upon treatment of WT neurons with LLOMe as compared to with vehicle (**E–H**: N = 3 experiments, Mann–Whitney test). **I–M** Damage to the endoplasmic reticulum does

not trigger the degradation of the negative regulators of autophagy. **I** Representative western blot from lysates of WT murine embryonic cortical neurons treated with vehicle (DMSO) or 50 nM Tunicamycin (TM) for 3 hrs. **J** Fold change in ATF4 levels upon treatment of WT neurons with TM as compared to with vehicle. **K** Fold change in MTMR5 levels upon treatment of WT neurons with TM as compared to with vehicle. **L** Fold change in MTMR2 levels upon treatment of WT neurons with TM as compared to with vehicle. **M** Fold change in Rubicon levels upon treatment of WT neurons with TM as compared to with vehicle (**I–M**: $N = 4$ experiments, Mann–Whitney test). **N–Q** MitoSR is not triggered by mitochondrial damage in astrocytes. **N** Representative western blot from lysates of WT murine astrocytes treated with vehicle (EtOH, DMSO) or 5 μM Ant A, 10 μM Oligo A for 6 hrs. **O** Fold change in MFN-2 levels upon treatment of WT astrocytes with 5 μM Ant A, 10 μM Oligo A as compared to with vehicle. **P** Fold change in MTMR2 levels upon treatment of WT astrocytes with 5 μM Ant A, 10 μM Oligo A as compared to with vehicle. **Q** Fold change in Rubicon levels upon treatment of WT astrocytes with 5 μM Ant A, 10 μM Oligo A as compared to with vehicle (**N–Q**: $N = 4$ experiments, Mann–Whitney test). Error bars indicate SEM. Source data are provided as a Source Data file.

(Fig. 6E). Following incubation at 37 °C, the neurons were imaged live using a confocal microscope. We observed a small but not statistically significant increase in the number of total lysosomes marked by LysoPrime Green in Rubicon knockdown neurons (Fig. 6F, G). More strikingly, we found a significant increase in the number of acidic lysosomes marked by pHLys Red upon Rubicon knockdown (Fig. 6F, H). We quantified the ratio of acidic to total lysosomes upon Rubicon knockdown and observed a significant increase in this ratio in neurons depleted of Rubicon (Fig. 6F, I). To extend this observation, we asked if the increased number of acidic lysosomes observed upon Rubicon knockdown represented a concomitant increase in the number of functional lysosomes. To test this, we knocked down Rubicon and measured the number of functional active lysosomes using Magic Red, a dye that fluoresces upon hydrolysis by the lysosomal protease Cathepsin B. We found an ~1.8-fold increase in the number of Magic Red puncta upon Rubicon knockdown (Fig. 6J, K), demonstrating a role for Rubicon in regulating lysosomal activity in neurons. Thus collectively, these results indicate that Rubicon influences lysosomal biogenesis and regulates lysosomal pH and degradative function in neurons.

We then asked how Rubicon's effect on lysosomal function impacts autophagosomal maturation in neurons. Within the neuronal soma, the majority of autophagosomes marked by LC3 are usually autophagolysosomes that are also positive for LAMP1[47,86]. Interestingly, however, we found that the majority of the lysosomes (marked by LAMP1-Halo) that are associated with EGFP-Rubicon do not colocalize with autophagosomes (marked by mCherry-LC3) (Fig. 6L, M). Hence, we questioned whether Rubicon blocks autophagosome-lysosome fusion in neurons. For this, we transfected neurons with either EGFP or EGFP-Rubicon along with mCherry-LC3 and LAMP1-Halo. We found no change in the number of mCherry-LC3 puncta per neuronal soma upon overexpression of Rubicon (Fig. 6N, O). However, when we measured colocalization between mCherry-LC3 and LAMP1-Halo as a readout of autophagolysosome formation, we found over-expression of EGFP-Rubicon led to an ~50% decrease in puncta positive for both mCherry-LC3- and LAMP1-Halo (Fig. 6N, P).

To determine whether this decrease in autolysosome number was due to reduced autophagosome-lysosome fusion or due to an inhibition of autophagosome biogenesis, we quantified the ratio of LAMP1+LC3+ autolysosomes to total LC3+ autophagosomes in EGFP or EGFP-Rubicon overexpressing neurons. In EGFP-overexpressing neurons, we found almost ~70% of LC3+ autophagosomes colocalized with lysosomes; this number decreased to ~34% in neurons overexpressing GFP-Rubicon, indicating an ~50% drop in autophagosome-lysosome fusion (Fig. 6N, Q). This finding correlates with our knockdown studies,

where we see an increase in autolysosome number upon Rubicon knockdown (Fig. 6A, D), collectively suggesting that removal of Rubicon triggers autophagic flux in neurons. In line with this, when we deplete Rubicon from neurons, we see reduced levels of Synapsin-1, one of the most abundant autophagic cargoes in neurons[12], suggesting increased turnover by autophagy (Supplemental S6F, G). Thus, our data indicate that Rubicon tightly regulates autophagy flux in neurons by influencing both lysosome biogenesis and function.

## Depletion of negative regulators promotes mitophagy flux in neurons

Previous work has shown that the myotubularin-related proteins MTMR5 and MTMR2 interact to suppress autophagosome formation in neurons[48], and here we find that Rubicon negatively regulates lysosomal function to modulate autophagosome-lysosome fusion. Thus, during acute mitochondrial stress when MitoSR is activated, the concerted degradation of MTMR5, MTMR2 and Rubicon is predicted to upregulate autophagic flux to promote the efficient turnover of damaged mitochondria. Hence, we asked whether depletion of these negative regulators would be sufficient to promote more efficient mitophagic flux in neurons undergoing mild oxidative damage at levels that are below the threshold to trigger MitoSR (Fig. 1A–3 nM Ant A). From a disease perspective, this might represent a powerful therapeutic approach, as it is thought that most neurological disorders where autophagy is compromised are chronic in nature, with the neurons undergoing low but sustained levels of mitochondrial stress over time. These levels may be insufficient to activate MitoSR, leading to an inefficient clearance of damaged mitochondria. To test this hypothesis, we nucleofected neurons with shRNA plasmids against *Rubicon* and the PI3P phosphatase *Mtmr2* at DIV = 0. After 6–7 days of knockdown (Supplemental S7A–C), we performed a mitophagy flux assay (Fig. 7A). In this assay, we labeled neurons with the mitophagy dye DMP Red, which primarily labels acidified mitochondria, indicating that they are engulfed within an acidified autolysosome (see methods)[87,88]. As an additional test, we used Lysotracker to label acidic lysosomes and checked for colocalization between the mitophagy dye and Lysotracker to quantify the number of mitophagolysosomes. Under basal conditions, consistent with mitochondria being a major cargo of neuronal autophagosomes[12], concerted removal of MTMR2 and Rubicon was able to promote increased mitochondrial turnover by autophagy as seen by increased colocalization of DMP Red and Lyso-tracker (Supplemental S7D, E). Next, we induced mild mitochondrial damage, in order to model chronic cases of neurodegenerative disease by treating neurons with 3 nM Ant A[36] for 1 hr, followed by a 2 hrs chase to probe for changes in mitophagy flux. Compared to neurons treated

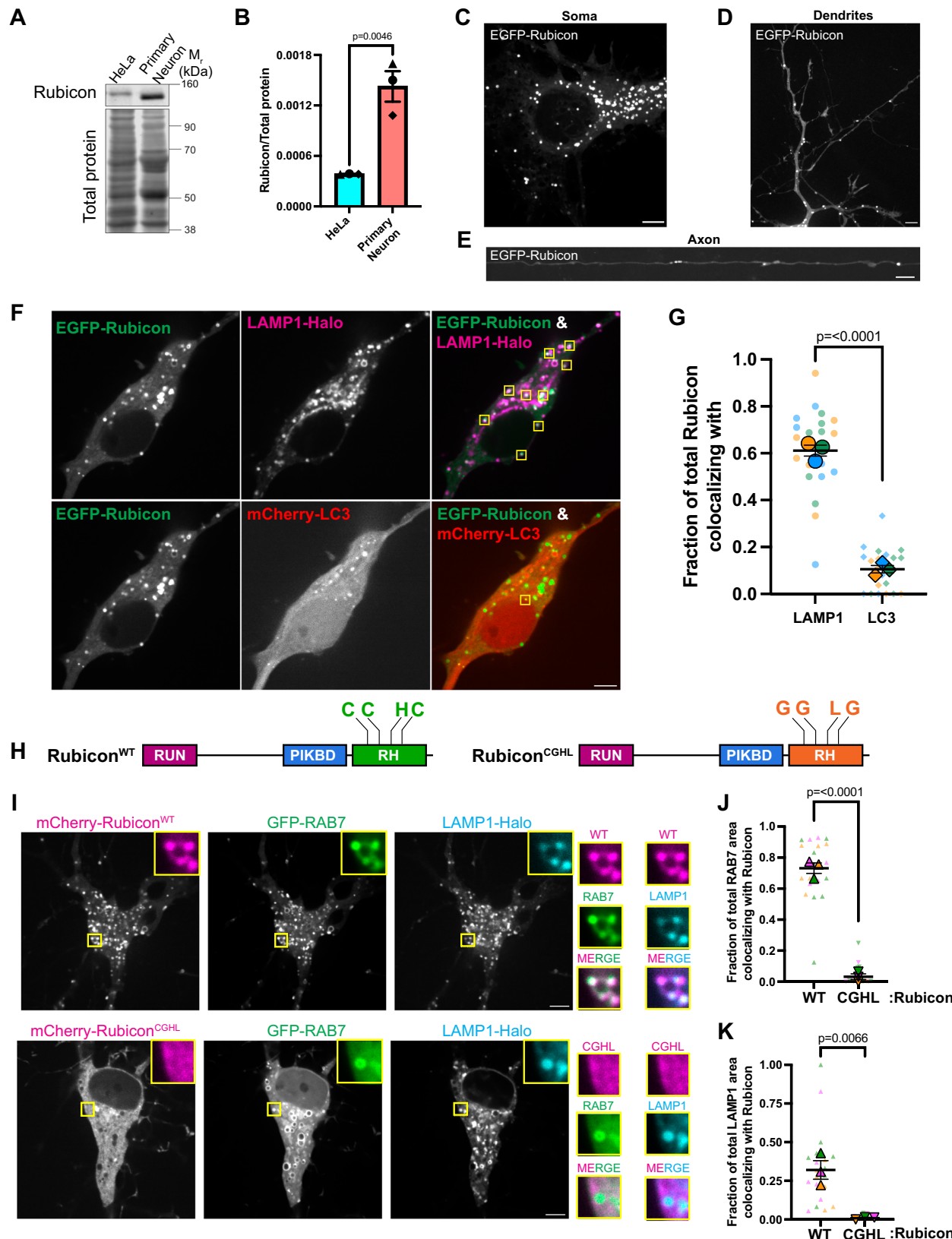

with a control shRNA, combinatorial knockdown of MTMR2 and Rubicon resulted in a significant increase in the number of mitophagolysosomes (Fig. 7B, C). The combined depletion of MTMR2 and Rubicon resulted in an approximately threefold increase in mitophagy flux in neurons undergoing low levels of mitochondrial stress. This observation suggests that the targeted depletion of the negative regulators of autophagy can result in enhanced mitochondrial turnover under low levels of oxidative stress.

Since MTMR2 affects the early steps of autophagosome biogenesis, and Rubicon predominantly the later steps of autophagosome maturation, we asked if these proteins act independently or in concert to repress mitophagy. To test this, we individually knocked down

**Fig. 5 | Rubicon's localization to lysosomes in neurons is RAB7-dependent.**
**A**, **B** Rubicon expression is enhanced in neurons. **A** Representative western blot from lysates of HeLa cells or WT murine embryonic cortical neurons and probed for Rubicon. **B** Rubicon band intensity normalized to total protein in HeLa cells or WT neurons ($N = 3$ experiments, two-tailed unpaired $t$ test). **C**–**E** Rubicon localizes to organelles distributed throughout the soma, dendrites, and axons of primary cortical neurons. **C** Representative max projection of the soma of a WT cortical neuron transfected with EGFP-Rubicon. **D** Representative max projection of the dendrites of a WT cortical neuron transfected with EGFP-Rubicon. **E** Representative max projection of an axon of a WT cortical neuron transfected with EGFP-Rubicon. **F**, **G** Rubicon colocalizes with the late endosome/lysosome marker LAMP1-Halo but not with the autophagosome marker mCherry-LC3. **F** Single z-plane confocal images of the soma of a WT cortical neuron transfected with mCherry-LC3, LAMP1-Halo and EGFP-Rubicon. Yellow boxes indicate EGFP-Rubicon colocalizing with either LAMP1-Halo or mCherry-LC3. **G** Fraction of the number of EGFP-Rubicon puncta in the soma of each neuron colocalizing with LAMP1-Halo or mCherry-LC3 ($N = 3$ experiments, two-tailed unpaired $t$ test). **H**–**K** Rubicon localization to lysosomes is RAB7-dependent. **H** Schematics of the domain organization of Rubicon$^{WT}$ and Rubicon$^{CGHL}$ with the RUN, PI3K-binding domain (PIKBD) and Rubicon homology (RH) domains annotated (domains not drawn to scale). **I** Single z-plane confocal images of somas of neurons transfected with GFP-RAB7, LAMP1-Halo and either mCherry-Rubicon$^{WT}$ or mCherry-Rubicon$^{CGHL}$. Yellow boxes indicate inset regions. **J** Fraction of total GFP-RAB7 area in each neuronal soma colocalizing with mCherry-Rubicon$^{WT}$ or mCherry-Rubicon$^{CGHL}$. **K** Fraction of total Halo-LAMP1 area in each neuronal soma colocalizing with mCherry-Rubicon$^{WT}$ or mCherry-Rubicon$^{CGHL}$ (**I**–**K**: $N = 3$ experiments, two-tailed unpaired $t$ test). All panels: Error bars indicate SEM, scale bars = 5μm. Source data are provided as a Source Data file.

MTMR2 (Supplemental S7F, G) and Rubicon (Supplemental S6A, B) in neurons and tested the impact on mitophagy flux upon mild mitochondrial damage (3 nM Ant A). We found that depletion of MTMR2 resulted in a modest increase in the number of mitophagolysosomes as compared to control (Fig. 7D, E). However, in neurons in which Rubicon was knocked down, we observed a significant increase in the number of mitophagolysosomes (Fig. 7D, E). When we quantified the change, we saw that knockdown of Rubicon resulted in ~2-fold increase in mitophagic flux (Fig. 7F). It is also interesting to note that the number of mitophagosomes does not change significantly between vehicle-treated and Ant A-treated control neurons with wild-type levels of Rubicon (Fig. 7D, E). This observation reinforces our finding that levels of negative regulators, especially Rubicon, are unchanged under mild oxidative stress (compare vehicle vs. 3 nM Ant A treatment in Fig. 1A), and these levels are sufficient to inhibit mitochondrial turnover. Additionally, our results demonstrate that depleting Rubicon had a greater impact on mitophagy flux than MTMR2 depletion (Fig. 7F), potentially due to the fact that MTMR2 acts early in the autophagy pathway while Rubicon predominantly affects the penultimate step by blocking autophagosome–lysosome fusion.

## Discussion

Mitochondrial dysfunction can have devastating effects on neuronal health and has been implicated in the pathogenesis of several neurological disorders, including PD or ALS[89]. The post-mitotic nature of neurons makes it essential that these cells either repair or degrade damaged mitochondria in order to prevent the release of ROS, Ca$^{2+}$, and/or mitochondrial DNA[89–91]. Here, we report the induction of Mitophagic Stress Response, or MitoSR, that is triggered upon high levels of mitochondrial damage in neurons. MitoSR operates in parallel to the canonical Pink1/Parkin-mediated mitophagy pathway (Fig. 8), triggering the degradation of Rubicon via the ubiquitin-proteasome system (UPS), and the degradation of MTMR5 and MTMR2 by both proteolysis and the UPS. These negative regulators actively suppress autophagic flux in distinct but complementary mechanisms, while targeted depletion of MTMR2 and Rubicon is sufficient to significantly stimulate neuronal mitophagy.

Damaged mitochondria have been shown to elicit a variety of stress response pathways. For instance, as a part of the well-characterized integrated stress response mechanism, the damaged organelle can attenuate translation of mitochondrial proteins by initiating the phosphorylation and hence deactivation of the translation elongation factor eIF2α[92]. Alternatively, misfolded proteins within mitochondria can induce the transcriptional activation of mitochondrial proteases and chaperones and ROS detoxification enzymes to initiate repair processes through a pathway known as mitochondrial unfolded protein response (UPR$^{mt}$)[93]. Mitochondrial dysfunction can also inhibit protein import into the organelle, triggering the proteasomal degradation of accumulated mitochondrial proteins in the cytosol in a pathway termed UPR activated by mistargeting of proteins (UPRam)[94]. While these mechanisms are initiated to repair damaged mitochondria to restore functionality, mitophagy is a degradative mechanism to eliminate toxic mitochondrial fragments that are irreversibly damaged. MitoSR is induced under conditions of acute mitochondrial stress and allows the neuron to more effectively degrade the damaged organelles. This response may be particularly critical in neurons because of the high metabolic activity and post-mitotic nature of this cell type.

Our results show that the degradation of negative regulators of autophagy by MitoSR is brought about predominantly by the proteasomal machinery, and not by autophagosomes. The proteasomal machinery has previously been reported to regulate levels of several autophagic proteins (WIPI2[95], ULK1[96], ATG16L[97]) under basal conditions. Consistent with this, we now find that Rubicon levels are regulated by the E3 ligase HECDT1 under basal levels. Independent of this, mitochondrial damage also triggers proteasomal degradation of several mitochondrial outer membrane proteins[98,99]. However, the activation of MitoSR represents a coordinated response that degrades multiple autophagy inhibitors to accelerate mitochondrial turnover under conditions of high oxidative stress. It is indeed striking that the MitoSR is specific for mitochondrial stress, as when we induced damage to lysosomes or the ER in neurons using LLOMe[76] or Tunicamycin[78], respectively, we did not detect degradation of these negative regulators. Additionally, the MitoSR is not activated by mitochondrial damage in either primary astrocytes or HeLa cells, suggesting that this activation may be a neuron-specific response to high levels of mitochondrial stress.

Along with the MTMR5-MTMR2 complex, which has previously been reported to suppress autophagosome biogenesis[48], here we identify Rubicon as a critical negative regulator of basal neuronal autophagy (Figs. 6 and 8). Rubicon is abundantly expressed in neurons and localizes to somal LAMP1-positive endosomes or lysosomes. In accordance with previous studies on Rubicon in non-neuronal systems[53,54,58–61], we find that in neurons, Rubicon is recruited in a RAB7-dependent manner to lysosomal membranes via its C-terminal RH domain. In contrast to recent findings in HeLa cells[61], we find that in neurons, Rubicon has a potent inhibitory effect on mitophagy through modulation of lysosomal physiology. When Rubicon is depleted, there is a significant enrichment in the number of acidic lysosomes and an even greater increase in the number of autolysosomes. In accordance with this, we observed increased lysosomal function (as measured by Cathepsin B activity) when Rubicon is knocked down in neurons. Thus, in neurons, Rubicon associates with lysosomes (Fig. 5F, G) and modulates lysosomal acidification (Fig. 6F, I) and thereby their hydrolytic activity (Fig. 6J, K). This inhibition directly impacts the latter steps of autophagy, primarily affecting autophagosome-lysosome fusion (Fig. 6N, Q). Thus, MTMR5/2 represses the early steps of autophagy[48], while Rubicon predominantly regulates the later steps, both of which are critical regulatory nodes in the context of neurological disorders where autophagy is compromised[100].

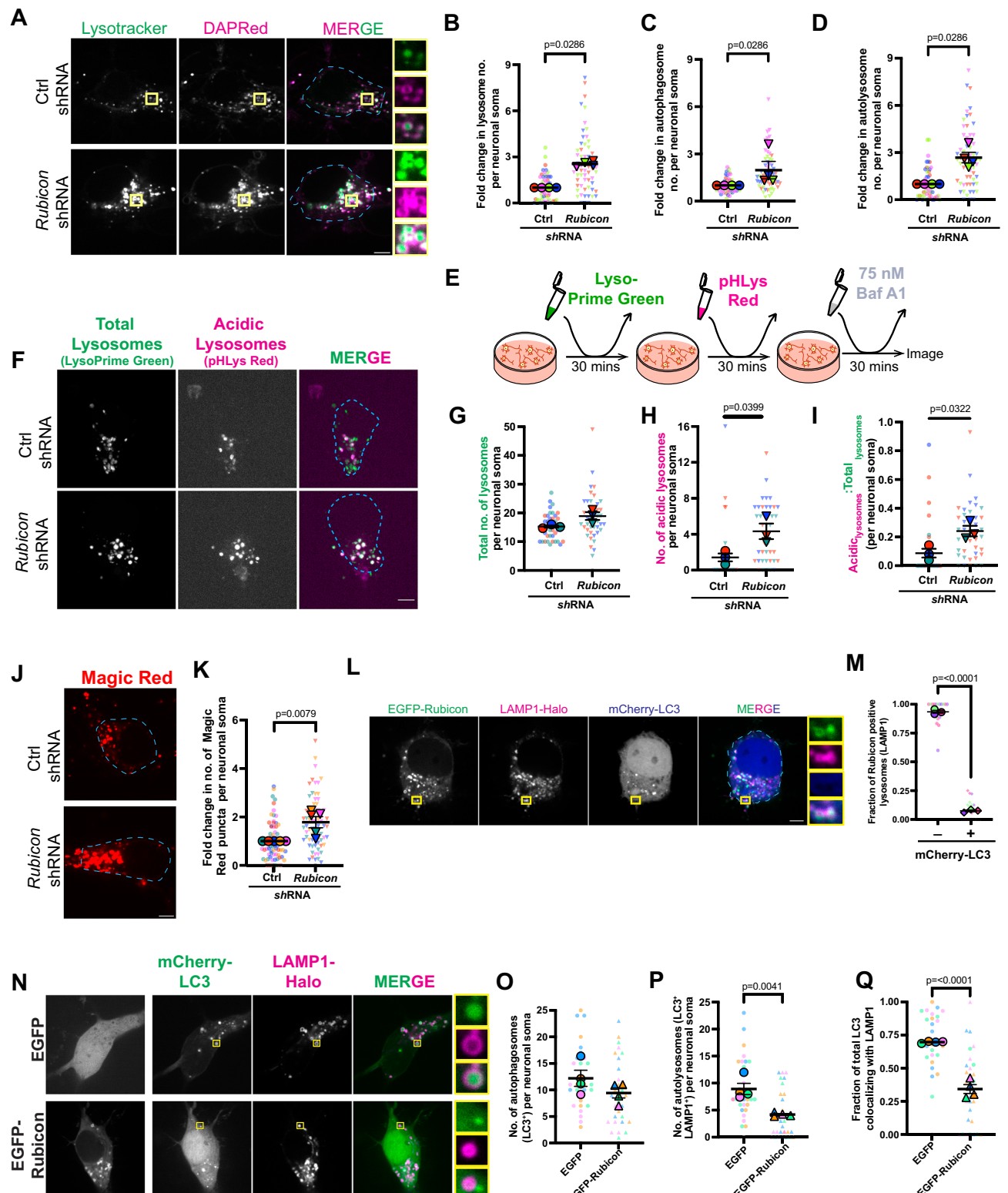

Previous work from our lab has shown that fusion of mitophagosomes with lysosomes is slow, and in fact is the rate-limiting step in neuronal mitophagy[36]. Here, we provide a mechanistic basis for this delay in fusion. At low levels of damage (3 nM), Rubicon is not degraded (Fig. 1A, D) and given its role in blocking autophagosome-lysosome fusion (Fig. 6N, Q), it hinders mitophagosome acidification by lysosomes. However, when the MitoSR is activated at higher levels of damage, the degradation of Rubicon facilitates the fusion of autophagosomes carrying damaged mitochondria with lysosomes.

Consistent with this hypothesis, when we deplete Rubicon levels in neurons and induce mild mitochondrial damage, we observe a twofold increase in mitophagy flux. Additionally, we also saw a modest increase in mitophagy flux upon depleting MTMR2. As a part of the MitoSR, knockdown of MTMR2 will upregulate autophagosome formation, but because Rubicon controls the fusion of these autophagosomes with lysosomes downstream, it acts as the limiting factor in regulating flux.

Our study opens interesting avenues for future investigation. Why do neurons require such tight negative regulation of autophagy? What

**Fig. 6 | Rubicon inhibits lysosomal function and autophagosomal maturation.**
**A–D** Rubicon knockdown increases the number of lysosomes and autolysosomes in cortical neurons. **A** Single z-plane confocal image of neurons nucleofected with control or *Rubicon* shRNA and treated with DAPRed and Lysotracker. Outline of the neuronal soma for quantification is indicated using dashed lines in cyan. Yellow boxes indicate inset regions. **B** Fold change in lysosome numbers (marked by Lysotracker punctae) per neuronal soma in control vs Rubicon knockdown neurons. **C** Fold change in autophagosome numbers (marked by DAPRed punctae) per neuronal soma in control vs Rubicon knockdown neurons. **D** Fold change in autolysosome numbers (marked by colocalizing DAPRed and Lysotracker punctae) per neuronal soma in control vs Rubicon knockdown neurons (**A–D**: $N = 4$ experiments, Mann–Whitney test). **E** Schematic depicting the steps of lysosomal pH detection assay in neurons using LysoPrime Green, pHLys Red to label total and acidic lysosomes, respectively, followed by treatment with 75 nM Baf A1.
**F–I** Rubicon negatively influences lysosomal acidification in neurons.
**F** Representative max projections of neurons nucleofected with Ctrl or *Rubicon* shRNA and assayed for lysosomal pH using the protocol represented in (**E**). Outline of the neuronal soma for quantification is indicated using dashed lines in cyan
**G** Total no. of lysosomes labeled by LysoPrime Green in control or Rubicon knockdown neurons. **H** No. of acidic lysosomes labeled by pHLys Red in control or Rubicon knockdown neurons. **I** Ratio of acidic to total lysosomes in control or Rubicon knockdown neurons (**F–I**: $N = 3$ experiments, two-tailed unpaired *t* test).

**J, K** Rubicon knockdown increases the number of proteolytically active lysosomes. **J** Single z-plane confocal images of neurons nucleofected with control or *Rubicon* shRNA and treated with Magic Red dye. Outline of the neuronal soma for quantification is indicated using dashed lines in cyan. **K** Fold change in the number of Magic Red puncta per neuronal soma in control vs Rubicon knockdown neurons ($N = 5$ experiments, Mann–Whitney test). (**L, M**) Rubicon is associated with lysosomes but not autolysosomes. **L** Single z-plane confocal images of a neuron transfected with EGFP-Rubicon, LAMP1-Halo and mCherry-LC3. Outline of the neuronal soma for quantification is indicated using dashed lines in cyan. Yellow boxes indicate inset regions. **M** Fraction of lysosomes (LAMP1-Halo) colocalizing with EGFP-Rubicon in the soma that do or do not colocalize with mCherry-LC3 ($N = 3$ experiments, two-tailed unpaired *t* test). **N–Q** Overexpression of Rubicon inhibits autophagosome maturation. **N** Single z-plane confocal images of neurons transfected with mCherry-LC3, LAMP1-Halo and either EGFP or EGFP-Rubicon. Yellow boxes indicate inset regions. **O** Number of autophagosomes (marked by mCherry-LC3 punctae) per neuronal soma, either expressing EGFP or EGFP-Rubicon. **P** Number of autolysosomes (marked by colocalizing mCherry-LC3 and LAMP1-Halo puncta) per neuronal soma, either expressing EGFP or EGFP-Rubicon. **Q** Fraction of autolysosomes (LAMP1$^+$LC3$^+$) to autophagosomes (LC3$^+$) per neuronal soma either expressing EGFP or EGFP-Rubicon (**N–Q**: $N = 4$ experiments, two-tailed unpaired *t* test). All panels: Error bars indicate SEM, scale bars = 5μm. Source data are provided as a Source Data file.

happens if autophagy is upregulated in neurons either acutely or chronically? An interesting study in proximal tubular epithelial cells showed that in the absence of Rubicon, lysosomal function is overwhelmed[101]. There is excess transfer of phospholipids from different organellar membranes to lysosomes via autophagy, leading to the formation of expanded hyperactive lysosomes. Our work suggests that in neurons, Rubicon also acts as a critical checkpoint to regulate lysosomal function. By regulating lysosomal physiology, Rubicon may control basal autophagy flux to prevent dysregulated breakdown of cellular components by autolysosomes. Inhibiting autophagosome maturation or lysosomal function has been reported to trigger the secretion of autophagic cargo by extracellular vesicles[102,103], in a process described as 'secretory autophagy during lysosome inhibition (SALI)'. Given the role of Rubicon in regulating lysosomal function and later steps of autophagy, it will be interesting to explore the role of this protein in secretory autophagy.

In summary, here we identify multiple stress response mechanisms, collectively termed MitoSR, which are activated under acute mitochondrial damage to increase mitophagy by degrading the negative regulators of autophagy. We uncover a mechanistic role of Rubicon in repressing neuronal autophagy by regulating lysosomal function. Further, we show that under mild mitochondrial stress, the targeted depletion of these negative regulators can promote autophagic turnover of damaged mitochondria. This finding may be of particular importance during chronic neurodegenerative disease, where mitochondrial stress may be mild but continues to build over time. We thus propose that in degenerating neurons of patients with PD or ALS, where mitophagy is compromised, targeting negative regulators of autophagy may enhance mitochondrial turnover and prevent the pathological accumulation of dysfunctional mitochondria that is characteristic of these diseases.

## Methods
### Ethics statement
This study was performed in accordance with the Guide for the Care and Use of Laboratory Animals of the National Institutes of Health (NIH). All experiments using animals followed protocols approved by the Institutional Animal Care and Use Committee at the University of Pennsylvania.

### Chemicals and reagents
The following chemicals were used: Ethanol (Decan Labs, #2716), Dimethyl sulfoxide (DMSO, Sigma Aldrich, #D2650), Antimycin A (Sigma Aldrich, #A8674), Bafilomycin A1 (Sigma, Aldrich, #SML1661), MG132 (Selleckchem, #S2619), Oligomycin A (Sigma Aldrich, #75351), CCCP (Sigma Aldrich #C2759), LLoMe (Cayman Chemicals #16008), PYR 41 (Tocris Bioscience #2978), EGTA (RPI #E14100), BAPTA (Invitrogen™ #B6769), Tunicamycin (Sigma Aldrich #T7765). The following reagents were used for in vitro cultures: Poly-L-Lysine (Sigma Aldrich #P1274), Poly-D-Lysine (Gibco #A3890401), Horse Serum (Gibco #16050122), Fetal Bovine Serum (Sigma Aldrich #F2442), Sodium Pyruvate (Gibco #11360070), D-(+)-Glucose solution (Sigma Aldrich #G8769), Sodium Chloride (NaCl, Fisher BioReagents #BP358-1), Minimal Essential Medium (MEM, Gibco #11095072), GlutaMAX (Gibco #35050061), Penicillin-Streptomycin (10,000 U/mL, Gibco #15140122), Neurobasal Medium (Gibco # 21103049), B27 Supplement-serum free (Gibco #17504044), Hibernate® E Low Fluorescence (TransnetyxYX Tissue #HELF-500 ml), AraC (Sigma Aldrich, #C6645), Dulbecco's Modified Eagle's Medium (DMEM, Corning, #10-017-CM), DMEM-high glucose & pyruvate (Gibco #11995073).

### Experimental model and subject details
Mouse lines used in this study include wild type [C57BL/6 J (RRID: IMSR JAX:000664)] and Parkin$^{-/-}$ [B6.129S4-Prkntm1Shn/J (RRID: IMSR JAX:006582)]. Cortices were dissected out from mouse embryos at day 15.5 in accordance with protocols approved by the Institutional Animal Care and Use Committee at the University of Pennsylvania. Neurons were isolated by digestion with 0.25% Trypsin followed by trituration. The neuronal suspension was then plated in attachment media (MEM and supplements including 10% horse serum, 1 mM sodium pyruvate, 33 mM D-glucose and 37.5 mM NaCl) on poly-L-Lysine-coated plates. Post attachment (5–6 hrs), the media was replaced with maintenance media (MM) [Neurobasal and supplements including 33 mM D-glucose, 2 mM GlutaMax, 37.5 mM NaCl, Penicillin (100 units/mL)/Streptomycin (100 mg/mL), and 2% B27]. On the following day, 5 μM of AraC was added to restrict the growth of non-neuronal cells such as glia. For experiments involving knockdown with shRNA plasmid, 6–7 days in vitro (DIV) neurons were used. Other experiments were performed with DIV 8–10 neurons.

The protocol has been adapted from: https://doi.org/10.17504/protocols.io.81wgby723vpk/v1

Primary murine astrocytes were isolated based on previously published protocols[82].

The protocol has been adapted from: https://doi.org/10.17504/protocols.io.e6nvwdp9wlmk/v1

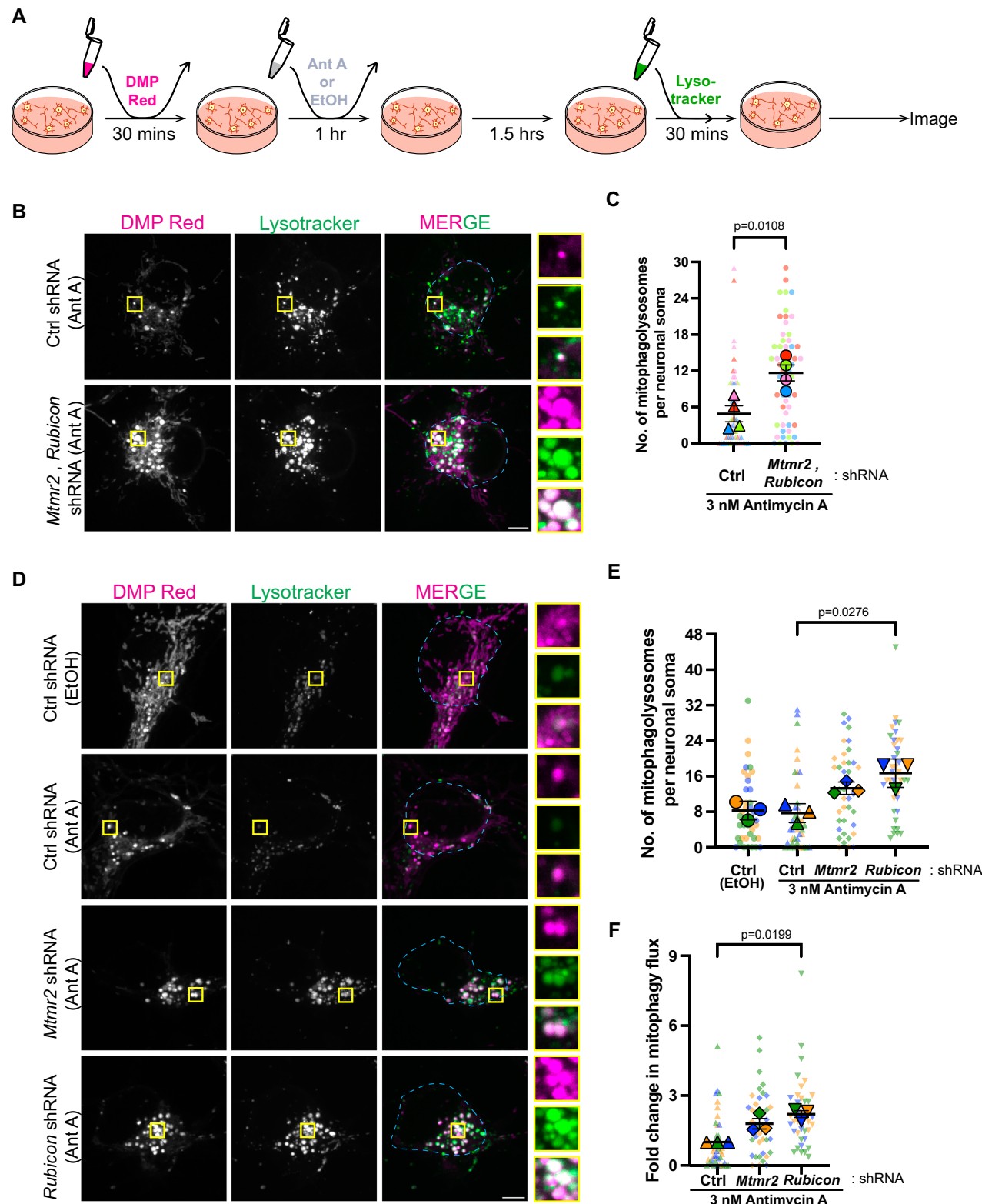

HeLa-M cells (referred to as HeLa in the main text, a gift from Andrew Peden, Cambridge Institute for Medical Research, UK) were authenticated and checked for mycoplasma contamination by STR profiling and MycoAlert detection kit (Lonza, LT07), respectively. Cells were grown in Dulbecco's Modified Eagle's Medium and supplemented with 1% GlutaMax and 10% Fetal Bovine Serum.

Neurons, astrocytes, and HeLa cells were grown in separate incubators at 37 °C with 5% $CO_2$.

### Transfection

Mouse embryonic cortical neurons were transfected using Lipofecta-mine 2000 (ThermoFisher Scientific Cat#11668019). A total of 1.5 μg DNA

**Fig. 7 | Depletion of MTMR2 and Rubicon promotes autophagic clearance of damaged mitochondria. A** Schematic depicting the steps of the mitophagy assay using DMP Red and Lysotracker to label acidified mitochondria and lysosomes, respectively. **B, C** Depletion of MTMR2 and Rubicon increases the number of mitochondria engulfed within autophagosomes. **B** Representative max projections of neurons nucleofected with Ctrl or *Mtmr2* and *Rubicon* shRNA and assayed for mitophagy using the protocol represented in (**A**). Outline of the neuronal soma for quantification is indicated using dashed lines in cyan. Yellow boxes indicate inset regions. **C** Number of mitophagolysosomes per soma (marked by colocalizing DMP Red and Lysotracker punctae) of neurons nucleofected with Ctrl or *Mtmr2* and *Rubicon* shRNA and assayed for mitophagy using the protocol represented in **A** (*N* = 4 experiments, two-tailed unpaired t-test). **D–F** Depletion of Rubicon is

sufficient to significantly enhance mitophagic flux. **D** Representative max projections of neurons nucleofected with Ctrl, *Mtmr2* or *Rubicon* shRNA and assayed for mitophagy using the protocol represented in (**A**). Outline of the neuronal soma for quantification is indicated using dashed lines in cyan. Yellow boxes indicate inset regions. **E** Number of mitophagolysosomes per soma (marked by colocalizing DMP Red and Lysotracker punctae) of neurons nucleofected with Ctrl, *Mtmr2* or *Rubicon* shRNA and assayed for mitophagy using the protocol represented in (**A**). **F** Fold change in mitophagy flux per soma of neurons nucleofected with Ctrl, *Mtmr2* or *Rubicon* shRNA and assayed for mitophagy using the protocol represented in (**A**) (**D–F**: *N* = 3 experiments, Kruskal–Wallis test). All panels: Error bars indicate SEM, scale bars = 5 µm. Source data are provided as a Source Data file.

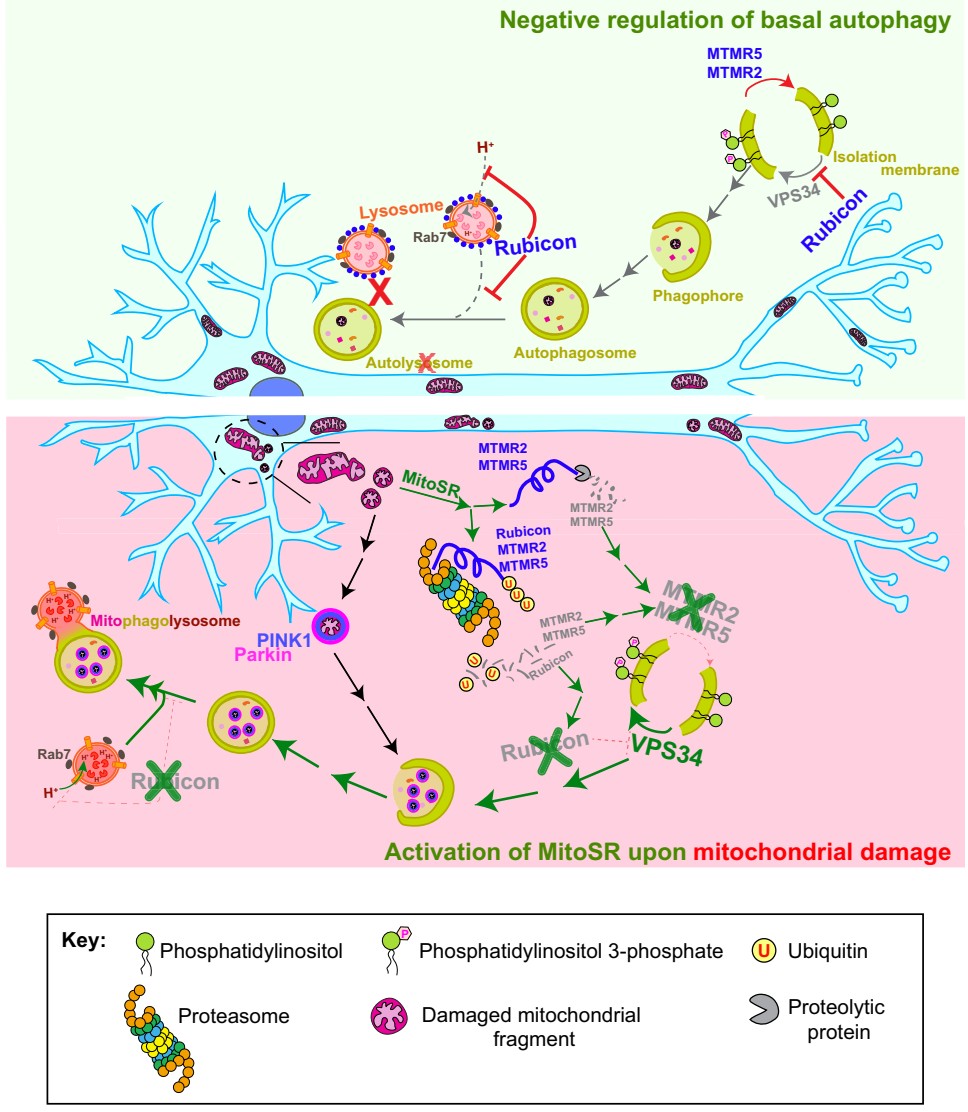

**Fig. 8 | Model: MTMR5/2 and Rubicon suppress neuronal autophagy under basal conditions, while acute mitochondrial stress induces the selective degradation of these negative regulators via MitoSR.** The upper panel depicts the roles of MTMR5/2 and Rubicon in repressing neuronal autophagy under basal conditions. MTMR5/2 activity hydrolyzes PI3P into PI, inhibiting early steps of autophagosome biogenesis[48]. Rubicon blocks lysosomal acidification and autophagosome-lysosome fusion, thereby impacting the latter steps of autophagy.

The lower panel depicts induction of MitoSR, which acts in parallel to the Pink1/Parkin pathway for mitophagy initiation in response to mitochondrial damage in neurons. Activation of MitoSR induces the ubiquitination of MTMR5, MTMR2 and Rubicon and the targeting of these negative regulators to the proteasome for degradation. MTMR5 and MTMR2 are also subject to proteolytic degradation. This concerted degradation facilitates increased autophagic flux, promoting mitochondrial turnover.

was mixed with 2 µl Lipofectamine and incubated for 30 mins at room temperature (RT) to form DNA-lipid complexes. The mixture was then added to neurons in fresh MM and incubated for 90 mins, post which the media was replaced with 1:1 conditioned: fresh MM. Neurons were

transfected for 18-24 hrs prior to imaging. The following plasmids were used for transfections: EGFP-Rubicon (RRID: Addgene_221659), LAMP1-Halo (RRID: Addgene_221655), mCherry-LC3 (RRID: Addgene_221656), mCherry-Rubicon^WT (RRID: Addgene_221657), mCherry-Rubicon^CGHL

(RRID: Addgene_221658) and EGFP-Rab7A (RRID: Addgene_28047), EGFP-N1 (RRID: Addgene_6085-1). HeLa cells were transfected with 1.5 μg of untagged human Parkin (RRID: Addgene_187897) using Lipofectamine 2000 for 18–24 hrs prior to treatment.

The protocol has been adapted from: https://doi.org/10.17504/protocols.io.81wgby723vpk/v1

## Nucleofection

On the day of dissection, post-trypsinization, embryonic neurons in suspension were nucleofected with shRNA plasmids. Nucleofections were carried out in an Amaxa nucleofector II device from Lonza according to the manufacturer's protocol. For each $10^6$ neurons, 3 μg of shRNA plasmid was used for nucleofection. The plasmids used were: Non-Target shRNA Control Plasmid DNA (Sigma Aldrich; SHC016), TRC mouse *Rubicon* shRNA (Sigma Aldrich; TRCN0000283271), TRC mouse *Mtmr2* shRNA (Sigma Aldrich; TRCN0000030094)

The protocol has been adapted from: https://doi.org/10.17504/protocols.io.kqdg3wo9zv25/v1

## Western blotting

Samples were lysed in RIPA buffer (50 mM Tris-HCl, 150 mM NaCl, 0.1% Triton X-100, 0.5% sodium deoxycholate and 0.1% SDS, pH = 7.4) supplemented with Halt Protease and phosphatase inhibitor (ThermoFisher Scientific Cat#78442) at 4 °C for 30 mins. Samples were then centrifuged at $17,000 \times g$ and the supernatant was collected. Protein concentration from the supernatant was measured using Pierce BCA Protein Assay Kit (ThermoFisher Scientific, Cat#23225). Samples were then denatured at 95 °C and resolved on an SDS-PAGE gel. The resolved proteins on the gel were then transferred to Immobilon-FL PVDF membranes (Millipore) at 100 V for 1 hr 12 mins. Post transfer, membranes were dried for 1 hr at RT and then rehydrated using methanol. The membranes were then stained with Revert™ 700 Total Protein Stain (Licor Cat#926-11021) for 5 mins and washed twice in water containing 6.7% acetic acid and 30% methanol, and then imaged in Odyssey CLx Infrared Imaging System (Li-COR). After this, membranes were destained with 0.1 M NaOH containing 30% methanol. Membranes were blocked for 5 mins in EveryBlot Blocking Buffer (Bio-Rad Cat#12010020) and then incubated with the appropriate primary antibody overnight at 4 °C. Membranes were washed three times the following day with 1× TBS (50 mM Tris-HCl, 274 mM NaCl, 9 mM KCl) with 0.1% Tween-20 (TBST). Membranes were incubated with the corresponding secondary antibody for 1 hr at RT, washed thrice in TBST and then imaged in Odyssey CLx Infrared Imaging System. If membranes had to be stripped, they were done in 1× NewBlot™ IR Stripping Buffer (Licor Cat# 928-40028) for 20 mins at RT and re-probed again starting from the blocking step. Band intensities were quantified in Image Studio™ Software (Li-COR). The following primary antibodies were used in this study: Rubicon at 1:1000 (CST D9F7; RRID: AB_10891617), MTMR2 at 1:2000 (sc-365184; RRID: AB_10708283), MTMR5 at 1:500 (sc-393488; RRID: AB_3097714), Parkin at 1:1000 (CST 2132; RRID: AB_10693040), VPS34 at 1:2000 (NB110-87320SS; RRID: AB_1199455), Mitofusin-2 at 1:1000 (sc-100560; RRID: AB_2235195), ATG7 at 1:1000 (ab133528; RRID: AB_2532126), ATG5 at 1:1000 (ab108327; RRID: AB_2650499), GAPDH at 1:1000 (ab9484; RRID: AB_307274), Actin at 1:1000 (Sigma Aldrich MAB1501R; RRID: AB_2223041), Tubulin at 1:2000 (CST 2148; RRID: AB_2288042), LC3B at 1:1000 (ab48394; RRID: AB_881433), Ubiquitin at 1:1000 (sc-8017; RRID: AB_628423), LAMP1 at 1:1000 (DSHB 1D4B, RRID: AB_2134500), Mitofilin at 1:1000 (ab110329, RRID: AB_10859613), HECTD1 at 1:2000 (ab101992, RRID: AB_10711075), ATF4 at 1:1000 (CST 11815, RRID:AB_2616025), SCARB2 at 1:1000 (LSBio LS-B305, RRID:AB_2182974), Synapsin-1 at 1:2000 (Sigma AB1543P, RRID:AB_90757), RAB7 at 1:1000 (ab50533, RRID:AB_882241), BNIP3 at 1:500 (sc56167, RRID:AB_2066767), TBK-1 at 1:1000 (CST 3013, RRID:AB_2199749), COX2 at 1:2000 (ab198286, AB_2861364), GABARAP at 1:1000 (ab109364;

RRID:AB_10861928), Cathepsin B at 1:1000 (CST 31718, RRID:AB_2687580) ATP6V1E1 at 1:1000 (Sigma GW22284F, RRID: AB_1845192), MTM1 at 1:500 (Proteintech 13924-1-AP, RRID:AB_2147700), MTMR14 at 1:500 (Proteintech 14973-1-AP RRID:AB_2147828). The following secondary antibodies were used at 1:20,000 dilution: IRDye® 800CW Donkey anti-Mouse IgG (Li-cor #926-32212, RRID: AB_621847); IRDye® 680RD Donkey anti-Mouse IgG (Li-cor # 926-68072, RRID: AB_1095362); IRDye® 800CW Donkey anti-Rabbit IgG (Li-cor # 926-32213, RRID: AB_621848); IRDye® 680RD Donkey anti-Rabbit IgG (Li-cor # 926-68073, RRID: AB_10954442), IRDye® 680RD Donkey anti-Chicken Secondary Antibody (Li-cor #926-68075, RRID:AB_10974977); IRDye® 800CW Goat anti-Rat IgG Secondary Antibody(Li-cor # 926-32219, RRID: AB_1850025)

The protocol has been adapted from: https://doi.org/10.17504/protocols.io.5jyl8j5zrg2w/v1

## Assays for autophagy, lysosomal function and mitophagy

To assay for autophagy, neurons were incubated with 50 nM DAP Red (Dojindo, D677-10) in fresh MM for 30 mins at 37 °C. The culture medium was aspirated out, and the neurons were washed with fresh MM. Neurons were then incubated with fresh MM containing 50 nM Lysotracker Green (ThermoFisher Scientific #L7526) for another 25–30 min at 37 °C and imaged live in imaging media containing 50 nM Lysotracker Green.

A detailed protocol is available at: https://doi.org/10.17504/protocols.io.36wgqqkz3gk5/v1

To test lysosomal function, neurons were incubated with Magic Red Substrate (MR-RR2) (Biorad, ICT937) at a final dilution of 1:4000 in fresh MM for 15 mins at 37 °C. Post incubation, neurons were imaged live in imaging media containing the Magic Red substrate at 1:4000 dilution.

A detailed protocol is available at: https://doi.org/10.17504/protocols.io.5jyl8q437l2w/v1

To assay for mitophagy, neurons were incubated with 20 nM Mtphagy dye (Dojindo, MD01-10; labeled here as DMP Red) in fresh MM for 30 mins at 37 °C. The culture medium was aspirated out, and the neurons were washed twice with fresh MM. Neurons were then incubated with EtOH or 3 nM Ant A in MM for 1 hr and then incubated for additional 1.5 hrs in fresh MM without EtOH or Ant A. Lysotracker Green (ThermoFisher Scientific #L7526) at a final concentration of 50 nM was added, and the neurons were then incubated for a final 25–30 mins. The media was then aspirated out and imaged live in imaging media containing 50 nM Lysotracker Green.

A detailed protocol is available at: https://doi.org/10.17504/protocols.io.j8nlkrpj6v5r/v1

## Assay for lysosomal pH

Lysosomal pH was detected using the Lysosomal Acidic pH detection kit from Dojindo (L266). Neurons were first incubated with LysoPrime Green in fresh MM for 30 mins at 37 °C at a final dilution of 1:20,000. The culture medium was aspirated out, and the neurons were washed twice with fresh MM. Neurons were then incubated with pHLys Red for 30 mins at 37 °C at a final dilution of 1:10,000. The culture medium was again aspirated out, and the neurons were washed twice with fresh MM. Neurons were imaged to check for proper labeling of both dyes. Post confirmation, imaging media was aspirated out and neurons were then incubated with 75 nM Baf A1 in fresh MM for 30 mins for partial lysosomal deacidification. Post incubation, Baf A1-containing media was aspirated out, and neurons were imaged live in imaging media.

A detailed protocol is available at: https://doi.org/10.17504/protocols.io.kqdg3wr47v25/v1

## Immunoprecipitation of Rubicon and MTMR2

Neurons were treated with ethanol or 30 nM Ant A for 2 hrs. Post incubation, neurons were washed with 1XPBS and then lysed in lysis buffer containing 50 mM HEPES (pH=7.4), 1 mM EDTA, 1 mM MgCl₂,

25 mM NaCl, 0.5% Triton X-100, 20 µg/mL Leupeptin, 2 mM DTT, 20 µg/mL TAME, 2 µg/mL Pepstatin A, 1 mM PMSF, along with 200 nM of the deubiquitinase inhibitor N-ethylmaleimide. Post lysis of the neurons, the lysate was spun at $17,000 \times g$ for 10 mins, and the pellet was discarded. 5% of the supernatant was stored as input to be used for western blotting later. The remaining supernatant was divided into two equal parts- to one part either 6 µg of Rubicon Antibody (CST E5J5V, RRID: AB_3697738) or 6 µg of MTMR2 (sc-365184) antibody was added; and to other part 6 µg of the corresponding IgG control [for Rubicon- Rabbit IgG (CST 2729S, RRID: AB_1031062) and for MTMR2- mouse IgG (Vector labs I-2000-1, RRID: AB_3668950)] was added. Supernatants were incubated overnight with the antibody or its corresponding IgG at 4 °C in an orbital rotor. The following day, 50 µl of Dynabeads Protein A (for Rubicon Ab and Rabbit IgG) (Invitrogen 10002D) or Dynabeads Protein G (for MTMR2 Ab and mouse IgG) (Invitrogen 10004D) per sample was washed thrice with 1× PBS containing 0.02% Tween-20 followed by one final wash in the lysis buffer. After washing, the beads were incubated with the antibody or IgG-containing supernatants for 4 hrs at 4 °C in an orbital rotor. Post incubation, the beads containing the IP-ed complex were magnetically isolated and washed four times with 1× PBS. IPed proteins were finally eluted in 2× SDS-containing sample buffer at 95 °C for 10 min, and subsequently used for western blotting.

A detailed protocol is available at: https://doi.org/10.17504/protocols.io.4r3l21m7pg1y/v1

### Ubiquitin enrichment assay

Neurons were treated with 10 µM MG132 for an additional 1 hr prior to adding 15 nM Ant A and incubating it for 2 hrs. Post incubation, neurons were washed with 1× PBS and lysed in lysis buffer containing 50 mM HEPES (pH=7.4), 1 mM EDTA, 1 mM MgCl₂, 25 mM NaCl, 0.5% Triton X-100, 20 µg/mL Leupeptin, 2 mM DTT, 20 µg/mL TAME, 2 µg/mL Pepstatin A, and 1 mM PMSF. Deubiquitination inhibitors provided with Ubiquitin Detection Kit (Cytoskeleton, BK161) were also added to the lysis buffer at 1× concentration. Post lysis of the neurons, lysates were spun at $17,000 \times g$ for 10 mins, and the pellet was discarded. 5% of the supernatant was stored as input to be used for western blotting later. The assay was performed using the Ubiquitin Detection Kit (Cytoskeleton, BK161), following the steps as mentioned in the manufacturer's protocol, with some modifications. Briefly, ubiquitination affinity beads and ubiquitination IP control beads were washed with 1× PBST twice and 1× PBS once, and then incubated with equal volumes of the lysate for 4 hrs at 4 °C. Post incubation, beads were centrifuged at $5000 \times g$ for 4 °C for 1 min. Beads were washed thrice with BlastR-2™ wash buffer and centrifuged. After final wash and centrifugation, beads were incubated with Bead Elution Buffer for 5 mins at RT and then centrifuged in a spin column at $10,000 \times g$ for 1 min to collect the immunoprecipitates. 2-mercaptoethanol was added to each sample, which was then boiled at 95 °C for 10 min to be used for western blotting.

A detailed protocol is available at: https://doi.org/10.17504/protocols.io.eq2ly4z7rlx9/v1

### Live imaging and image analysis

When required, prior to imaging, neurons were labeled with 100 nM of Janelia Fluor 646 Halo Ligand (Promega, Cat: #GA112A) for 20 mins followed by a washout for another 20–30 mins. Neurons were imaged in imaging media containing HibernateE Brain Bits and supplemented with 33 mM D-glucose, 37.5 mM NaCl and 2% B27. Neurons were imaged live inside an environmental chamber at 37 °C for 45–60 mins. Imaging was performed on a PerkinElmer UltraView Vox spinning disk confocal on a Nikon Eclipse Ti Microscope with an Apochromat 100x 1.49 N.A. oil-immersion objective and a Hamamatsu CMOS ORCA Fusion (C11440-20UP) camera with VisiView (Visitron).

To quantify colocalization between EGFP-Rubicon, LAMP1-Halo and mCherry-LC3 in the soma, punctae were identified manually from a single z-plane across channels and binarized. The number of colocalizing puncta was then calculated using the AND function in Fiji/ImageJ. For colocalization experiments with GFP-Rab7, LAMP-Halo and mCherry-Rubicon^WT or mCherry-Rubicon^CGHL, cell areas from a single z-plane were thresholded and binarized using Yen thresholding in Fiji/ImageJ. Area of colocalization was then calculated using the AND function in Fiji/ImageJ.

To quantify DAPRed, Lysotracker numbers from a single z-plane, the region of interest, i.e., the soma, was identified. DAPRed and Lysotracker positive puncta were then segmented using Ilastik segmentation tool, a machine learning segmentation software. Images were binarized, and the numbers calculated using the Analyze Particles option in Fiji/ImageJ. The number of colocalizing DAPRed and Lysotracker puncta was calculated using the AND function.

To quantify the number of Magic Red positive puncta from a single z-plane, the region of interest, i.e., the soma, was identified. Magic Red positive puncta were then segmented using Ilastik segmentation. Images were binarized, and the numbers calculated using the Analyze Particles option in Fiji/ImageJ. To measure lysosomal acidification using LysoPrime Green and pHLys Red, max projections of each channel from each image were generated. LysoPrime Green and pHLys Red positive puncta from the soma were then segmented using Ilastik. Images were binarized and the numbers calculated using the AND and the Analyze Particles functions in Fiji/ImageJ.

To measure mitophagolysosome nos. (i.e., colocalizing DMP Red and Lysotracker puncta), max projections of each channel from each image were generated. DMP Red and Lysotracker positive puncta from the soma were then segmented using Ilastik. Images were binarized, and the no. of colocalizing DMP Red and Lysotracker puncta was calculated using the AND function.

A detailed protocol is available at: https://doi.org/10.17504/protocols.io.261gek6eog47/v1

### Immunostaining

MM was aspirated out, and neurons were washed once with 1× PBS. Neurons were then fixed with 4% paraformaldehyde supplemented with 4% sucrose (4% PFA/4% sucrose) for 10 mins at RT. Post fixation, samples were washed with PBS thrice, and then permeabilized with ice-cold methanol for 8 mins at −20 °C. Post permeabilization, neurons were washed and blocked in blocking solution (5% goat serum and 1% BSA in PBS) for 90 mins at RT. Neurons were then incubated with the primary antibody in the blocking solution overnight at 4°C. The following day, post 14–16 hrs incubation, neurons were washed thrice with 1× PBS and incubated with secondary antibodies, also in blocking solution, for 60 minutes at RT. Post incubation, samples were washed, mounted in ProLong Gold, and imaged. The following primary antibodies were used: p62 at 1:250 (ab56416, RRID: AB_945626), LAMP1 at 1:100 (DSHB 1D4B, RRID: AB_2134500), MAP2 at 1:500 (Sigma Aldrich AB5622, RRID: AB_91939). The following secondary antibodies were used at 1:1000 dilution- Alexa Fluor 488 Goat anti-Rat IgG (H + L) (A11006, RRID: AB_2534074), Alexa Fluor 555 Goat anti-Rabbit IgG (H + L) (A21429, RRID: AB_2535850), Alexa Fluor 633 Goat anti-Mouse IgG (H + L) (A21052, RRID:AB_2535719).

The protocol has been adapted from: https://doi.org/10.17504/protocols.io.rm7vzqpb8vx1/v1

### Lentiviral transduction

Neurons at DIV = 1 were transduced with lentiviral particles (MOI = 9) obtained from Horizon Dharmacon™ reagents. Transduction was confirmed by visualization of GFP signal post 3–4 days after infection. At DIV 6–8, MM containing virus was removed, and neurons were treated with ethanol or 30 nM Ant A for 2 hrs. Post-treatment samples were lysed and prepared for western blotting. pSMARTEF1a/TurboGFP plasmid was used by Dharmacon for preparing the lentivirus with

shRNA sequence against *Hectd¹* (Cat#V3SM7593-236146236) or non-targeting control (Cat #S03-005000-01).

A detailed protocol is available at: https://doi.org/10.17504/protocols.io.rm7vz9mb4gx1/v1

## Statistics & reproducibility

All statistical analyses were performed on Graphpad Prism Version 10.4.1. All column graphs and superplots were generated on Graphpad Prism Version 10.4.1. Data points from all biological replicates were tested for normal distribution using the Shapiro-Wilk normality test. Each data point on a column graph represents one biological replicate. For superplots, each biological replicate has been represented by colored shapes with black outlines, and its corresponding technical replicates are represented with the same translucent color and shape. All quantitative analysis was performed on at least three biological replicates per condition. Data are represented as mean, and error bars indicate the standard error of the mean. $P < 0.05$, $P < 0.01$, $P < 0.001$, and $P < 0.0001$ were considered significant. The sex of the murine embryos was not determined during neuron isolation. No statistical method was used to predetermine sample size. Neurons from multiple embryos from a mouse were pooled together and randomly assigned to any experimental condition. Neurons that looked unhealthy and showed signs of blebbing were excluded from image acquisition. Images of poor quality were removed during analysis. The investigators were not blinded to allocation during experiments and outcome assessment.

## Reporting summary

Further information on research design is available in the Nature Portfolio Reporting Summary linked to this article.

## Data availability

All raw files associated with this study have been uploaded on Zenodo. Information on all lab materials used in this study has been provided in the Key Resource Table (KRT) file uploaded on Zenodo. All new protocols have been uploaded on protocols.io. Source data are provided with this paper.

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

## Acknowledgements

We thank Julia F. Riley for sharing astrocyte samples, and Mariko Tokito and Karen Jahn for technical assistance. We thank Dr. Dan A. Tudorica and Prof. James H. Hurley (UC, Berkeley) for insightful discussions and for gifting us the EGFP-Rubicon clone. We are very grateful to Dr. Sierra Palumbos for extensive feedback on the manuscript. We also thank Dr. Elizabeth R. Gallagher, Dr. Juliet Goldsmith, Dr. Kaya Matson, Dr. Yewon Jeon, and all other members of the Holzbaur laboratory for insightful discussions. We are grateful to Dr. Dorotea Fracchiolla for her help with project management. This research was funded in its entirety by Aligning Science Across Parkinson's (ASAP-000350) through the Michael J. Fox Foundation for Parkinson's Research (MJFF) to E.L.F.H. For the purpose of open access, the authors have applied for a CC-BY 4.0 public copyright license for all Author Accepted Manuscripts arising from this submission.

## Author contributions

B.B. and E.L.F.H. designed research and wrote the paper, and B.B. performed research and analyzed data.

## Competing interests

The authors declare no competing interests.
