## [Transparent Peer Review file · Nature Communications]

Mitochondrial damage triggers the concerted degradation of negative regulators of neuronal autophagy

Corresponding Author: Professor Erika Holzbaur

Version 0:

Reviewer comments:

Reviewer #1

(Remarks to the Author)

The authors have carried out a tremendous amount of work and have very carefully and elegantly addressed the majority of my comments and concerns. In particular, they have addressed all of my most crucial points, with the exception of the identity of the E3 ligase(s), but I concede that at this stage, that would be out of scope for this already data-heavy paper. I congratulate the authors on their efforts and believe that Nature Communications would be appropriate for this paper.

Reviewer #2

(Remarks to the Author)

The authors have addressed the concerns raised, performed new experiments that allow their interpretations to be strengthened and also toned down the claims.

made.

Response to Review

Basak and Holzbaur

Mitochondrial damage triggers concerted degradation of negative regulators of neuronal autophagy

Editorial concerns

As you will see, although the reviewers found the work interesting, they raised serious concerns that question the degree of conceptual advance that the findings represent over previous work, and the strength of the data and of the novel conclusions that can be drawn at this stage. In particular, the reviewers had concerns about how the data fit with existing studies; they were critical of the lack of additional insight into the mechanism (e.g., which E3 ligase(s) may be involved) and requested further characterizations of the full response and its specificity to mitochondria.

Thank you for the thoughtful feedback, which has guided us in the thorough revision of this work. We have addressed each of the concerns raised with additional supportive data as described in detail below, and have also restructured and clarified our presentation to more clearly position our new findings in the context of the current understanding of mitophagy in neurons.

Importantly, in this work we have identified a mitochondrial-stress specific response mechanism that directly regulates the autophagic turnover of damaged mitochondria in neurons. Modulating autophagy by conventional methods has been challenging in neurons. Hence, our work identifying a neuronal stress response pathway in which Rubicon acts as a critical regulator of mitophagic flux in response to oxidative damage is an important advance. These findings offer fresh insights into the cell biology of neuronal stress responses, and new insights that may be relevant to the development of therapeutic approaches to Parkinson's and ALS.

Since we agreed that the work would be significantly strengthened by the inclusion of additional mechanistic information and a more complete demonstration of the specificity of the response, we have worked hard to fill in the gaps identified by the reviewers in the first round. We agree that the addition of these data significantly enhances the impact of the work. Thus, it is our hope that you will find this fully revised manuscript to merit publication in *Nature Communications*

Reviewer #1 (Remarks to the Author):

The paper by Basak and Holzbaur describes a pathway parallel to PINK-Parkin, modulating mitophagy in neurons, which is mediated by the degradation of negative autophagy regulators MTMR5, MTMR2, and Rubicon. The authors use several techniques, including immunoblotting and fluorescent microscopy imaging to support their hypothesis. They show that proteasome-mediated degradation of these three proteins is induced by mitochondrial damage in murine neurons, which in turn de-represses discrete steps in autophagic turnover of mitochondria. Moreover, the authors show that this process is parkin-independent, pointing at the MitoSR being a novel pathway facilitating mitophagy in neurons. The authors also show that Rubicon localizes to lysosomes via interaction with Rab7 and blocks autolysosome formation, as well as reducing lysosome acidification and lysosomal proteolytic function. Finally, the authors show that Rubicon has a stronger effect on mitophagy than MTMR2, given its involvement at a more downstream step in the pathway.

Overall, while this manuscript presents a number of novel and interesting findings, there remain a several important issues related to the scope of the advance, the interpretation of the data and the conclusions being drawn from the findings.

We thank the reviewer for this careful summary of our work, and their appreciation of the novel and interesting findings we have made. We also agree that thorough responses to the points raised has made this work stronger and clearer.

With respect to the scope of the advance, there are three major items that would help increase the significance and impact of the manuscript. First, while the authors argue for a new pathway (MitoSR) and show the stress-induced degradation of three key autophagy inhibitor proteins by the proteasome pathway, the mechanism remains elusive. While they exclude Parkin, no other E3 ligase, including those involved in autophagy or mitochondrial quality control, were tested. No general unbiased approach (i.e. CRISPR-based screen of E3s) to identify the E3 was attempted.

We agree that identifying the relevant E3 ligase is of strong interest, and took a candidate-based approach focusing on the E3 ligase HECTD1, recently reported to ubiquitinate and regulate Rubicon levels via the proteasome in human chondrocytes (Liao et al., 2023).

We used shRNA to deplete HECTD1 in primary cortical neurons and observed a 47% increase in Rubicon levels (**Response Fig. 1**). Consistent with our data that Rubicon levels increase in neurons treated with the proteasome inhibitor MG132 (Fig 2 A, D), this result suggests that the E3 ligase HECTD1 regulates Rubicon levels basally via the ubiquitin proteasomal machinery.

Response Figure 1: (A) Representative western blot from lysates of wild type cortical neurons transduced with lentivirus containing shRNA against HECTD1 or non-targeting control, and treated with vehicle (EtOH). (B) Quantification of fold change in HECTD1 levels in neurons transduced with lentivirus containing shRNA against HECTD1 or non-targeting control. (C) Quantification of fold change in Rubicon levels in neurons transduced with lentivirus containing shRNA against HECTD1 or non-targeting control. (N=4 experiments, Mann-Whitney test, * $p < 0.05$, error bars indicate S.E.M)

However, in parallel experiments in which shRNA-treated neurons were incubated with Antimycin A, we observed a significant knockdown of HECTD1 but only a marginal (non-significant) increase in Rubicon levels (**Response Fig. 2**). These data are now included in **Supplemental Figure S4 (A-F)** in the revised manuscript

Response Figure 2: (A) Representative western blot from lysates of wild type cortical neurons transduced with lentivirus containing shRNA against HECTD1 or non-targeting control, and treated 30 nM Ant A for 2 hrs. (B) Quantification of fold change in HECTD1 levels in neurons transduced with lentivirus containing shRNA against HECTD1 or non-targeting control, and treated with Ant A. (C) Quantification of fold change in Rubicon levels in neurons transduced with lentivirus containing shRNA against HECTD1 or non-targeting control, and treated with Ant A. (N=4 experiments, Mann-Whitney test, * $p < 0.05$, ** $p < 0.01$, error bars indicate S.E.M)

Of note, we see that HECTD1, which is primarily cytosolic in neurons under basal conditions, is recruited to damaged mitochondria upon Ant A treatment in a dose dependent fashion (**Response Fig. 3**).

Response Figure 3: (A) Representative images of wild type neurons treated with EtOH or 3 and 15 nM Ant A and immunostained for Mitofilin and HECTD1. (B) Quantification of area of colocalization of HECTD1 with Mitofilin in wild type neurons treated with EtOH or 3 and 15 nM Ant A (N=4 experiments, One way ANOVA, *p<0.05, ** p<0.01, *** p<0.001 error bars indicate S.E.M, scale bar= 5 μm)

These findings establish a connection between HECTD1 and mitochondrial damage, but suggest that HECTD1 may no longer be appropriately localized to degrade lysosomally-associated Rubicon upon mitochondrial insult. Thus, it is likely that additional E3 ligases contribute to the regulation of Rubicon levels upon mitochondrial stress. HECTD1 is a member of the HECT family of 28 E3 ligases, and it is possible other members of the HECTD1 family may compensate for loss of HECTD1 function. Indeed, the concept of multiple E3 ligases regulating protein levels during mitochondrial stress has been investigated before. For instance, in the absence of Parkin, the E3 ligase MUL1 has been reported to regulate levels of the outer mitochondrial membrane protein Mitofusin-2 to facilitate mitophagy (Yun et al., 2014). Similarly, HUWE1, which is also a member of the HECT family of proteins is known to ubiquitinate Mitofusin-2 upon mitochondrial damage to facilitate mitophagy (Di Rita et al., 2018). Given the strong localization of HECTD1 to damaged mitochondria, and our data supporting a role for HECTD1 in the ubiquitination of Rubicon under basal conditions, we can say that HECTD1 is likely to be one of the E3 ligases involved in this stress response mechanism. Unfortunately, there are ~600 E3 ligases and the functional redundancy among these enzymes makes CRISPR-based screening of E3 ligases particularly challenging in primary murine neurons. Since our data suggest that this stress response is specific to neurons, considerable work will be required to more fully address this point.

Second, the authors propose that the pathway is specific for mitochondrial stress but only one other stressor is tested and not very convincingly (see comments below RE: LLOMe, etc...). Since mitochondrial specificity is a major conclusion of the paper, this seems to be a gap. For example, induction of general autophagy (Rapamycin, starvation...), ER stress (Thapsigargin, tunicamycin,...), over-expression of misfolded proteins, just to name a few, should have been tested.

We now provide additional data showing that the Mitophagic Stress Response described here is specific to induction of mitochondrial damage. Other forms of cellular stress are unable to induce degradation of the negative regulators of autophagy.

As noted in the manuscript **Figure 4E-H**, LLoMe treatment of cortical neurons did not induce the selective degradation of MTMR2, MTMR5, and Rubicon. But the reviewer was concerned that we had not included data indicating that lysophagy was successfully induced in these experiments. As shown in **Response Fig. 4**, we see clear evidence of the recruitment of the lysophagy adaptor SQSTM1/p62 to damaged lysosomes (Gallagher and Holzbaur, 2023) in cortical neurons under the conditions used for this assay.

Response Figure 4: Representative max projections of neurons treated with vehicle (EtOH) or 1 mM LLoMe for 2 hrs, fixed and stained for p62 (magenta), LAMP1 (green) and the neuronal marker MAP2 (blue). Outline of the soma is marked. Scale bar = 5 μ m. (The experiment was repeated thrice, and representative images from one biological replicate are shown.)

Next, we tested whether the induction of ER stress by tunicamycin in neurons led to the degradation of negative regulators of autophagy. As shown in **Response Fig. 5**, Tunicamycin treatment induced ER stress as confirmed by elevated levels of ATF4 (Hoyer et al., 2024), but did not induce the degradation of MTMR5, MTMR2, or Rubicon. These data are now included in **Figure 4 (I-M)** of the revised manuscript.

Response Figure 5: (A) Representative western blot from lysates of WT murine embryonic cortical neurons treated with vehicle (DMSO) or 50 nM Tunicamycin (TM) for 3 hrs. (B) Fold change in ATF4 levels upon treatment of WT neurons with TM as compared to with vehicle. (C) Fold change in MTMR5 levels upon treatment of WT neurons with TM as compared to with vehicle. (D) Fold change in MTMR2 levels upon treatment of WT neurons with TM as compared to with vehicle. (E) Fold change in Rubicon levels upon treatment of WT neurons with TM as compared to with vehicle (N=4 experiments, Mann-Whitney test, ns=not significant, * p <0.05, error bars indicate S.E.M.).

Inhibition of proteasome activity can induce proteotoxic stress (Kandel et al., 2024) and in line with this, studies have indicated that MG132 can induce the formation of protein aggregates in cells (Prosser et al., 2022; Hao et al., 2023). Similarly, we find that proteotoxic stress can be induced by blocking proteasomal activity using MG132 in murine cortical neurons, as confirmed by induction of elevated levels of Ubiquitin (**Response Fig. 6**). However, this treatment does not result in changes in

degradation of the negative regulators of autophagy: MTMR5, MTMR2, and Rubicon (**Response Fig. 6**). These data are now included in **Supplemental figure S5 (B-G)** in the revised manuscript.

Response Figure 6: (A) Representative western blots from lysates of WT murine embryonic cortical neurons treated with either veh (vehicle-EtOH) or 10 μ M MG132 for 3 hours, and probed for ubiquitin (Ub) (B) Fold change in Ub levels upon treatment of neurons with 10 μ M MG132 as compared to with vehicle. (C) Representative western blots from lysates of WT murine embryonic cortical neurons treated with either veh or MG132 for 3 hours. (D) Fold change in MTMR5 levels upon treatment of neurons with MG132 as compared to with vehicle. (E) Fold change in MTMR2 levels upon treatment of neurons with MG132 as compared to with vehicle. (F) Fold change in Rubicon levels upon treatment of neurons with MG132 as compared to with vehicle. (N=4 experiments, Mann-Whitney test, ns=not significant, * p <0.05, error bars indicate S.E.M).

Finally, in previous work we have shown that neither starvation nor mTOR inhibition leads to a significant upregulation of autophagy in primary rodent neurons (Maday and Holzbaur, 2016). A similar observation was made in human iPSC-derived neurons by Chua et al. (2022), so we would not expect to see significant degradation of negative regulators of autophagy under these conditions.

Third, the authors test whether the paradigm that activates MitoSR (AntiA) also leads to degradation of other autophagy proteins (Supplemental Figure 4). However, they focus on three positive regulators (or effectors) of autophagy (VPS34, ATG7, ATG5), whereas the logic of their pathway would suggest that more congruent targets would involve the degradation of autophagy inhibitors. It seems that here a more general and unbiased approach such as SILAC (or other quantitative mass spec approach, +/- AntiA and Proteasome inhibitor) would be warranted.

To address this point, we have now examined the effects of mitophagy induction (using 30 nM Ant A) on: mitophagy-related proteins (MFN-2, Parkin, TBK1, RAB7, BNIP3), proteins involved in the positive regulation of autophagy (VPS34, ATG5, ATG7, GABARAP, LC3), as well as lysosomal proteins (Cathepsin B, SCARB2, LAMP1, ATPV1E1), other members of the MTMR5/2 family (MTMR1,

MTMR14), and housekeeping proteins (actin, GAPDH); please see **Response Figure 7** and **Figure 3** in the revised manuscript. Across these experiments, we do not see significant changes in the levels of most of proteins tested, except for MFN-2 (indicative of initiation of mitophagy upon mitochondrial damage) and Cathepsin B (indicative of increased lysosomal protease activity possibly due to Rubicon degradation). These observations are fully consistent with recent proteomic screens from the Muqit lab, which also demonstrate that the induction of mitochondrial stress does not induce a generalized turnover of cellular proteins (Antico et al., 2021); instead, the response we are seeing is much more specific.

Response Fig. 7. (A) Representative western blot of wild type cortical neurons treated with vehicle (EtOH) or 30 nM Ant A for 2 hrs and probed for MFN-2 (B) Quantification of MFN-2 levels in neurons treated with vehicle (EtOH) or Ant A. (C-H) Quantification of levels of mitophagy proteins (Parkin, TBK1, RAB7) and mitochondrial proteins (COX II, Mitofillin) in neurons treated with wild type cortical neurons treated with vehicle (EtOH) or 30 nM Ant A for 2 hrs. (I-M) Quantification of levels of autophagy associated proteins (VPS34, ATG, ATG7, GABARAP, LC3) in neurons treated with wild type cortical neurons treated with EtOH or 30 nM Ant A for 2 hrs. (N) Representative western blot of wild type cortical neurons treated with EtOH or 30 nM Ant A for 2 hrs and probed for Cathepsin B (O) Quantification of Cathepsin B levels in neurons treated with EtOH or Ant A. (P-R) Quantification of levels of other lysosomal proteins (SCARB2, LAMP1, ATP6V1E1) in neurons treated with wild type cortical neurons treated with vehicle (EtOH) or 30 nM Ant A for 2 hrs. (S, T) Quantification of levels of other MTM1 and MTMR14 in neurons treated with wild type cortical neurons treated with EtOH or 30 nM Ant A for 2 hrs. (U-W) Quantification of levels of house-keeping proteins (Tubulin, GAPDH, Actin) in neurons treated with wild type cortical neurons treated with EtOH or 30 nM Ant A for 2 hrs. (N=3 experiments, p<0.05, error bars indicate S.E.M., scale bars = 5 μ m.) Two-tailed unpaired t-tests were

performed to determine significance for most proteins, except for COX II, BNIP3, TBK-I. Mann-Whitney test was performed for TBK-1, COX II and BNIP3 to test for significance as the data were not found to be normal.

Other major Points:

- In Figure 1 and throughout the manuscript, the reduction of MTMR5 and Rubicon in response to AntiA are clear. However, the banding pattern of MTMR2 is much more suggestive of proteolytic cleavage. This possibility needs to be explored.

We addressed this point by testing the effects of three different broad-spectrum protease-inhibitors (Calpastatin, E64, and PD150606) at concentrations as high as 10 μ M for 3 hrs. However, none of these drugs blocked the degradation of MTMR2 in response to mitochondrial stress (**Response Fig. 8**), in contrast to the striking rescue seen with MG132 treatment (**Fig. 2 A, C**).

(Response Fig. 8: (A) Representative western blot from lysates of WT embryonic cortical neurons treated with vehicle (water, DMSO) or 10 μ M calpastatin (CalpS) or 10 μ M E64, 10 μ M PD150606 followed by an additional treatment with vehicle (EtOH) or 30 nM Ant A. **(B)** Ratio of intact MTMR2 normalized to total (intact + degraded) band intensities from WT neurons treated with vehicle (water, DMSO) or CalpS/E64/PD150606 followed by an additional treatment with vehicle (EtOH) or Ant A. (N=3 experiments, One way ANOVA with Sidak's multiple comparison test, ns= not significant, error bars indicate S.E.M).

- Related to the point above, in Figure 2, use of MG132 in the μ M concentration range is known to also inhibit calpains and cathepsins (which could in theory be responsible for the proteolytic cleavage of MTMR2). Thus the experiments with MG132 should be confirmed with another more specific proteasome inhibitor (i.e. Epoxomicin, Bortezomib, etc...).

We found that neither Epoxomicin (**Response Fig. 9A**) or Bortezomib (**Response Fig. 9B**) treatment was sufficient to induce the accumulation of ubiquitinated proteins in cortical neurons treated with AntA, indicating that neither of these drugs effectively inhibited the proteasome under the conditions of our assay. In contrast, treatment of murine cortical neurons with MG132 at concentrations well-established in the literature (Fulcher et al., *Nat Cell Bio* 2025; Aleci et. al., *Nat. Comm.* 2024; Prosser et. al. *Nat Cell Bio* 2022; Trulsson et al., *Nat Comm.* 2022) was sufficient to induce the pronounced accumulation of ubiquitinated proteins (please see **Response Fig. 9C**).

Response Fig. 9. (A) Representative western blots from lysates of WT cortical neuronal lysates treated with vehicle (DMSO) or 10 μ M of Epoxomicin (Epox) for 2 hrs followed by an additional treatment with vehicle (EtOH) or 30 nM Ant A for 2 hrs. Ubiquitin levels are not significantly altered between Ant A and Epox + Ant A treated conditions. (B) Representative western blots from lysates of WT cortical neuronal lysates treated with vehicle (DMSO) or increasing concentration of Bortezomib for 1 hr followed by an additional treatment with vehicle (EtOH) or 30 nM Ant A for 2 hrs. Ubiquitin levels are not significantly altered between Ant A and Ant A+ Bortezomib treated conditions. (C) Representative western blots from lysates of WT cortical neuronal lysates treated with vehicle (EtOH) or 10 μ M MG132 for 1 hr followed by an additional treatment with vehicle (EtOH) or 30 nM Ant A for 2 hrs. Ubiquitin levels are significantly altered between Ant A and Ant A+ MG132 treated conditions.

- In 2E, F: If the probed proteins are Ub-modified, why don't we see a ~8kd shift or laddering pattern of the three proteins on the blot? Is this really ubiquitylation? another orthogonal approach such IP followed by target and Ub blotting or, even better, mass spec would be important to confirm ubiquitination?

We now provide data showing that when we immunoprecipitate Rubicon from neurons treated with Ant A, we can detect a ubiquitinated Rubicon band migrating more slowly than Rubicon (**Response Fig. 10A** and **Figure 2 E** of the revised manuscript). We obtained similar results for MTMR2 (**Response Fig. 10B** and **Supplemental Figure S3A** of the revised manuscript). For MTMR5 we found the antibody ineffective for immunoprecipitation, hence we could not sufficiently enrich for low abundance ubiquitinated MTMR5.

Response Fig. 10: (A) Representative western blot from wild type cortical neuronal lysates treated with vehicle (EtOH) or 30 nM Ant A for 2 hrs and immunoprecipitated with Rubicon antibody or control Rabbit IgG. (B) Representative western blot from wild type cortical neuronal lysates treated with vehicle (EtOH) or 30 nM Ant A for 2 hrs and immunoprecipitated with MTMR2 antibody or control mouse IgG. (Experiments were performed 4 times, and images from one of the biological replicates are represented here)

- Moreover, in Figure 2E, F, a critical control is lacking. To show that degradation is stress-responsive, the authors need to compare the amount of pull down with Ub beads both with and without Anti A.

Our pull-down experiments using Ub-binding beads indicate that MTMR5, MTMR2 and Rubicon are ubiquitinated in response to Ant A treatment of cortical neurons. As a more direct test for ubiquitination increases in response to mitochondrial damage, we treated neurons with vehicle (ethanol) or Ant A (30 nM) and immunoprecipitated Rubicon and MTMR2 as shown above. We see a significant increase in the band corresponding to ubiquitinated Rubicon upon mitochondrial damage (**Response Figure 11A Figure 2 F** of the revised manuscript). This indicates that ubiquitination of Rubicon is increased during mitochondrial damage.

Response Fig. 11: A: Quantification of the ratio of ubiquitinated to total Rubicon upon immunoprecipitation of Rubicon in neurons treated with EtOH or Ant A. B Quantification of the ratio of ubiquitinated to intact and ubiquitinated MTMR2 upon immunoprecipitation of MTMR2 in neurons treated with EtOH or Ant A (N=4 experiments, two-tailed unpaired t-test, ns= not significant * $p < 0.05$, error bars indicate S.E.M).

For MTMR2 we also detected ubiquitin-positive bands that colocalize with MTMR2, but we did not see a significant increase in the ubiquitination of MTMR2 post mitochondrial damage (**Response Figure 11B** and **Supplemental Figure S3 B**). For MTMR5, the antibody failed to enrich for the protein in the immunoprecipitated fraction, hence ubiquitination could not be detected. But we are able to show:

- Pulldown of MTMR5 and MTMR2 by Ub-binding beads from neurons treated with Ant A and MG132 (**Figure 2G, I, J** of the revised manuscript).
- Blocking the activity of Ub-E1 enzymes with PYR41 (please see response figure 19 below and **Supplemental figure S3 C-F** in the revised manuscript) rescues the degradation of MTMR5 and MTMR2 induced by mitochondrial stress
- Blocking proteasomal degradation using MG132 rescues the degradation of Rubicon, MTMR5 and MTMR2 (**Figure 2 A-D** of the revised manuscript)

We thus also explored the possibility of non-proteasome-mediated degradation of MTMR5 and MTMR2. We observed that when we treated neurons with the calcium (Ca^{2+}) ethylene glycol-bis(β -aminoethyl ether)-N,N,N',N'-tetraacetic acid (EGTA), we observed complete rescue of intact MTMR5 and MTMR2 levels during mitochondrial stress (**Response Figure 12**). Since Ca^{2+} is a major regulators of mitochondrial homeostasis, it is likely that upon damage these ions play critical roles in activating specific proteins that degrade MTMR2 and MTMR5 to increase autophagy, and thereby restore mitochondrial homeostasis. Further work will be required to identify these pathways. These data have been included in **Supplemental Figure S3 G-I** of the revised manuscript

Response Fig 12: (A) Representative western blot from lysates of WT embryonic cortical neurons treated with vehicle (water, DMSO) or 5 mM EGTA or 20 μM BAPTA followed by an additional treatment with vehicle (EtOH) or 30 nM Ant A. (B) MTMR5 band intensity normalized to total protein from WT neurons treated with vehicle/ EGTA/ BAPTA and EtOH/Ant A. (C). Ratio of intact MTMR2 normalized to total (intact + degraded) band intensities from WT neurons treated with vehicle/ EGTA/ BAPTA and EtOH/Ant A (N=3 experiments, One way ANOVA with Sidak's multiple comparison test, ** $p < 0.01$, **** $p < 0.0001$, error bars indicate S.E.M.).

Based on these additional experimental data, we have revised our conclusions to state that activation of MitoSR activates the ubiquitin proteasomal machinery, in concert with additional degradative mechanisms, to selectively degrade negative regulators. To account for this complexity, we no longer refer to MitoSR as 'a pathway', but instead indicate that MitoSR includes the activation of coordinated stress response mechanisms that together target the negative regulators of autophagy in response to mitochondrial damage in neurons.

- Fig. 3I-L: Why are the authors using AA + Oligo in astrocytes but only AA in neurons? They should use AA alone as in Fig. 1., AA alone induces ROS but does not induce mitophagy, whereas AA + Oligo does, so they are not looking at the same paradigm. Similarly, these cells were treated for 6hrs to induce mitophagy whereas the previous experiments were for only 2hrs, at which time considerably less mitophagy has occurred. As the authors are making a strong claim about MitoSR being neuron-specific, the experiments done in astrocytes (and in HeLa cells – see below) need to be done rigorously under the same conditions.

Here, we must respectfully disagree with the reviewer's conclusion that Ant A treatment does not induce mitophagy, as concentrations of Antimycin A as low as 30 nM are sufficient to induce Mfn2 degradation in mouse cortical neurons as shown in **Response Figure 7**, indicating onset of PINK1/Parkin-

dependent mitophagy. Similarly, previous work has shown that concentrations of Ant A as low as 3 nM are sufficient to induce activation of PINK1/Parkin-dependent mitophagy, recruitment of the mitophagy receptor OPTN, and mitochondrial engulfment by LC3-positive autophagosomes in rat hippocampal neurons (Evans and Holzbaur, 2020; Harding et al., 2021).

We do agree that we should have been clearer in our initial submission that for all three cell types tested, we used conditions which we were confident would successfully induce mitophagy, based on previous studies (Wong and Holzbaur, 2014; Evans and Holzbaur, 2020; Harding et al., 2023). Importantly, we verified the induction of mitophagy in each model by monitoring the loss of Mitofusin-2 (Mfn2), a well-accepted early marker for activation of PINK1/Parkin-dependent mitophagy (McLelland et al., 2018; Chen & Dorn, 2013).

We also note that the conditions required to induce PINK1/Parkin-dependent mitophagy vary significantly across cell type. We find that the onset of PINK1/Parkin-dependent mitophagy is variable in response to 30 nM Antimycin A in either HeLa cells or astrocytes, but highly consistent in murine cortical neurons. However, to address the concern raised, we tested the cell-type specificity of MitoSR using 30 nM Ant A for 2 hrs in astrocytes **Response Figure 13** and HeLa cells **Response Figure 14**- conditions similar to what was used in neurons. Under these conditions, we did not see a decrease in the levels of MFN-2, indicating that mitophagy was not induced in these cells. Concomitantly, we did not see reduction in the levels of any of the negative regulators of mitophagy investigated here (**Response Figures 13, 14**)

Response Fig. 13. (A) Representative western blot from lysates of WT murine astrocytes treated with vehicle (EtOH) or 30 nM Ant A for 2 hrs. (B) Fold change in MFN-2 levels upon treatment of WT astrocytes with 30 nM Ant A as compared to with vehicle. (C) Fold change in Rubicon levels upon treatment of WT astrocytes with 30 nM Ant A as compared to with vehicle. (D) Fold change in MTMR2 levels upon treatment of WT astrocytes with 30 nM Ant A as compared to with vehicle. (N=3 experiments, Mann-Whitney test, ns=not significant, error bars indicate S.E.M).

HeLa + Parkin

Response Fig. 14. (A) Representative western blot from lysates of HeLa cells transfected with Parkin, and treated with vehicle (EtOH) or 30 nM Ant A for 2 hrs. (B) Fold change in MFN-2 levels upon treatment of Parkin transfected HeLa cells with 30 nM Ant A as compared to with vehicle. (C) Fold change in Rubicon levels upon treatment of Parkin transfected HeLa cells with 30 nM Ant A as compared to with vehicle. (D) Fold change in MTMR2 levels upon treatment of Parkin transfected HeLa cells with 30 nM Ant A as compared to with vehicle. (N=3 experiments, Mann-Whitney test, ns=not significant, error bars indicate S.E.M).

- The authors should compare WT to the Rubicon CGHL mutant in Figures 5A-F.

We attempted this experiment by nucleofecting neurons with shRNA to deplete Rubicon in conjunction with overexpression of Rubicon^{WT} or Rubicon^{CGHL}. Unfortunately, we consistently observed neuronal stress and cell death in these assays – see the representative images below (**Response Fig. 15**), so we cannot make this comparison. However, we refer the reader to previous work in HeLa cells, where expression of the Rubicon^{CGHL} mutant as compared to as wild type rescued autophagy flux (Bhargava et al., PNAS, 2020).

Response Fig. 15. Representative max projections of neurons nucleofected with *Rubicon* shRNA along with mCherry or mCherry-Rubicon^{WT} or mCherry-Rubicon^{CGHL}. Prolonged expression of Rubicon^{WT} or Rubicon^{CGHL} resulted in the formation of aggregated structures triggering neuronal stress.

- Strictly speaking, in the text corresponding to Figure 5C and F (and elsewhere), I don't think we can say "fusion" as the organelles may just be co-localized or tethered without actual fusion.

We have carefully reviewed the text to ensure that we use the term fusion only when warranted, and use the term co-localized when we cannot clearly conclude that fusion has occurred from confocal Z-stacks and/or tandem markers.

- In Figure 5G-J, if autophagic flux were increased by degradation of Rubicon as the authors propose, wouldn't one expect a reduction in autophagosomes/DAP Red staining since these organelles would be predicted to be degraded more rapidly by lysosomes? This should be clarified.

Our analysis of autophagosomal flux by western blots shows increased flux upon knockdown of Rubicon – we have added these additional data to our manuscripts (**Response Fig. 16** below and **Supplemental Figure S6 F, G** of the revised manuscript).

The DAP Red dye is incorporated into the autophagosome membrane and will remain fluorescent following autophagosome-lysosome fusion (<https://www.dojindo.com/products/D677/>); thus the dye does not specifically distinguish between autophagosomes and autolysosomes, so that the increase in DAP Red puncta upon Rubicon knockdown could be reflective of either enhanced autophagosome biogenesis or more autolysosomes. Indeed, we see more colocalizing DAP Red and LysoTracker Green puncta (please see **Figure 6 A, C** of the revised manuscript) in Rubicon knockdown neurons indicating more autolysosomal numbers.

- The imaging in Figures 4 and 5, showing various aspects of autophagosome-lysosomes is very nice. However, it would be important for the authors to show an orthogonal approach confirming the effects of the manipulating Rubicon (i.e. WT and Rubicon KO or Rubicon mutant cells) on autophagosome-lysosome interaction/flux (i.e. co-immuno-isolation or other biochemical approach).

As an orthogonal approach we tested the effect of Rubicon KD on the levels of the most abundant autophagic cargo in neurons - synapsin-1 (Goldsmith et al., 2022). We find that knockdown of Rubicon, leads to a 20% decrease in synapsin-1 levels, consistent with a significant activation of neuronal autophagy (**Response Fig. 16** and **Supplemental Figure S6 F, G** of the revised manuscript). In conjunction with previous work demonstrating that Rubicon represses autophagic flux in other cell types (Matsunaga et al., NCB, 2009; Zhong et al., NCB, 2009; Tabata et al., MBoC, 2010; Bhargava et al., PNAS, 2020) these data are consistent with the hypothesis that Rubicon impairs autophagic flux in neurons.

Response Figure 16: (A) Representative western blot from lysates of WT cortical neurons nucleofected with control or *Rubicon* shRNA plasmid, and probed for Synapsin-1. (B) Fold change in Synapsin-1 levels in neurons nucleofected with control or *Rubicon* shRNA plasmid (N= 6 experiments, *p<0.05, Mann-Whitney test, error bars indicate S.E.M.)

- In Figure 6B-C, as a control to monitor basal mitophagy, the authors should include conditions both with and without AntA in both genotypes.

As requested, we tested the effect of depleting Rubicon and MTMR2 on basal mitophagy, and saw an increase in the number of mitophagolysosomes, indicating that depletion of Rubicon and MTMR2 can

promote mitochondrial turnover by autophagy under both basal and stress conditions (**Response Fig. 17** below, and also see **Supplemental S7 D, E** in the revised manuscript).

Response Figure 17: Representative max projections of neurons nucleofected with Ctrl, *Mtmr2* and *Rubicon* shRNA and assayed for mitophagy using the protocol represented in manuscript Fig:7A. Outline of the neuronal soma for quantification is indicated using dashed lines in cyan. Yellow boxes indicate inset regions. (E) Number of mitophagolysosomes per soma (marked by colocalizing DMP Red and Lysotracker punctae) of neurons nucleofected with Ctrl, *Mtmr2* and *Rubicon* shRNA and assayed for mitophagy using the protocol represented in Fig:7A (N=4 experiments, ** $p < 0.01$, two-tailed unpaired t-test, error bars indicate S.E.M).

Minor Comments:

- In supplemental Figure 2, no effect of BafA1 or MG132 is seen in HeLa cells but this was not tested with AntiA, which is the condition that induces degradation in neurons (by comparison, the effect of MG132 on basal Rubicon is modest). Therefore, the authors should also test whether AntiA leads to MTMR2/5 and Rubicon in HeLa cells to more comprehensively explore neuronal specificity.

Since we observed no degradation of Rubicon or MTMR2/5 in HeLa cells treated with high doses of Ant A/Oligo A (Figure S5), it is unclear what might be learned from further analysis of the effects of either MGM132 or Bafilomycin A on this degradation.

- In supplemental Figure 3A, the effects of PYR41 would be expected to be very non-specific as it affects the E1, thereby disrupting almost all ubiquitylation. Concurrently, the effects shown are very modest, possibly owing to cell toxicity or some other non-specific effect. I'm not sure these data really add much to the paper.

While we agree that the data in supplemental Figure 3A provide limited insight into the specifics of the MitoSR response, these data do offer confirmatory support for the model that loss of Rubicon and MTMR5/2 is mediated by ubiquitination and proteasomal degradation rather than an alternative mechanism such calpain activation. Thus we feel it reasonable to include these data in the supplement but not the main text. In the revised submission, we also now include quantitative data showing that blocking the activity of all E1 enzymes using the PYR41 inhibitor during mitochondrial damage completely rescues the degradation of the negative regulators (**Response Figure 18** and Supplemental Figure S3 C-F).

Response Figure 18: (A) Representative western blots from lysates of WT murine embryonic cortical neurons treated with vehicle (DMSO) or 10 μ M PYR41 for 1 hr followed by an additional treatment with vehicle (EtOH) or 30 nM Ant A for 2 hrs. (B) Rubicon band intensity normalized to total protein from WT neurons treated with DMSO/PYR41 and EtOH/Ant A. (N=3 experiments, One way ANOVA with Sidak's multiple comparison test). (C). Ratio of intact MTMR2 normalized to total (intact + degraded) band intensities from WT neurons treated with DMSO/PYR41 and EtOH/Ant A (N=3 experiments, One way ANOVA with Sidak's multiple comparison test). (D) MTMR5 band intensity normalized to total protein from WT neurons treated with DMSO/PYR41 and EtOH/Ant A.

- In supplemental Figure 3B, the authors claim that LLoMe resulted in no change in the MitoSR proteins, despite lysosomal damage depicted by LC3-II level increase. This increase is very minimal, and visually not very convincing without quantification. To me, without more convincing evidence of lysosomal damage (beyond just LC3-II changes, for example increased Gal3 recruitment to lysosomes), these data seem overinterpreted. We are very happy to provide more detailed demonstration of the induction of lysophagy under these conditions, as described in our recent paper (Gallagher and Holzbaur, 2023). As shown below in **Response Fig. 19**, treatment of neurons with LLoMe under the conditions described, there is a specific recruitment of p62 aggregates to lysosomes marked by LAMP1. We have now replaced the LC3-II western data with our imaging data in the revised manuscript Supplemental Figure S5 A.

Response Figure 19: Representative max projection of neurons treated with vehicle (EtOH) or 1 mM LLoMe for 2 hrs, fixed and stained for p62 (magenta), LAMP1 (green) and the neuronal marker MAP2 (blue). Outline of the soma is marked. Scale bar= 5 μ m. (The experiment was repeated thrice, and representative images from one biological replicate are shown.)

- In supplemental Figure 4, the authors claim that VPS34 is not changed with Antimycin A treatment. However, the

quantification shows an increase (although not statistically significant). Moreover, there seems to be an increase in the lower band of ATG5. The authors should comment on these in the text.

Statistically we do not see any significant changes in VPS34 levels in neurons treated with 15 and 30 nM of Ant A. We do see an increase in the cleavage of ATG5 which has now been added to the text. Of note, based on feedback from the referees, we have now examined an increased number of proteins associated with autophagy, lysosomal function, and mitochondrial control (please see **Fig. 3** and **Supplemental Figure S4 G-K** of the revised manuscript). Statistical analyses of these data indicate significant changes are seen only for Mitofusion 2, a well-established marker for the induction of mitophagy; Rubicon, MTMR2, and MTMR5, the focus of this work; and Cathepsin B (an 70% decrease upon Ant A treatment). Given that we see an increase in Cathepsin B activity upon Ant A treatment, this decrease in Cathepsin B levels in neuronal lysates may reflect an increase in lysosomal degradative activity, and thus represent an indirect effect of the induction of Rubicon degradation.

- In supplemental Figure 4C-G, why not use 30nM of AA? It seems this is the most effective concentration in Figure 1.

Our quantitative analysis, shown in Figure 1, indicates there is no statistical difference between the response to either 15 or 30 nM Ant A, indicating that the higher value is not a more effective concentration. However, we also repeated the experiment using 30 nM Ant A, and saw no major changes in the levels of any of the proteins examined other than Mitofusin-2.

- At the bottom of page 9: should indicate HeLa cell “overexpressing Parkin” as the Parkin gene is deleted in HeLa cells.

Thank you for catching this – we have made the necessary correction.

- Figure craftsmanship: some lines in figures are crooked (fig 1E, under Antimycin a), some figure labels are not centered (Supplemental figure 4D).

The crooked lines in the pdf file supplied to referees were an unfortunate side-effect of the compressed figure resolution required to upload a review copy to the journal website. We will carefully review the final figures to ensure labeling is neat and centered.

Reviewer #2 (Remarks to the Author):

Basak and Holzbaur convincingly demonstrate that, specifically in neurons, acute mitochondrial inhibition results in the proteasome-dependent degradation of negative regulators of macroautophagy MTMR2/5 and Rubicon. The paper is generally rigorous and well-presented. This work adds to research on the differential regulation of autophagy in various cell types and will be of general interest to the field of cell biology, especially as it relates to neurodegenerative diseases and autophagy.

We thank this reviewer for their supportive and thoughtful comments.

However, the authors claim that Rubicon inhibits lysosome acidification and consequent lysosomal proteolysis (Figures 5 and 7) is not fully supported by the data. LysoTracker dye the authors employ accumulates in acidic compartments, but its fluorescence is fairly insensitive to changes in lysosome pH, so a greater number of LysoTracker positive structures might simply reflect an increase in the number of lysosomes. Similarly, without a sense of the rate at which Magic Red is converted to its fluorescent form by cathepsins, it's unclear whether the increased numbers of Magic Red puncta in Rubicon knockdown cells is simply an increase in the number of lysosomes. Given the weeklong timeframe of their Rubicon knockdown and overexpression experiments, an alternative explanation is that Rubicon regulates lysosome biogenesis. The authors could choose to modify or soften their conclusions or perform the following set of experiments to better validate their model:

- A more appropriate way to measure lysosome acidity would be to use a more pH-sensitive probe, such as LysoSensor (ThermoFisher) or pHLyso Red (Dojindo).

- Assess accumulation of fluorescent Magic Red in a time-course experiment in neurons to better establish changes in cathepsin activity.

We very much appreciate these suggestions about examining changes in lysosome biogenesis or activity upon Rubicon depletion. Supporting this possibility, we have new data showing that levels of the lysosomal proteins LAMP1 and SCARB2 increase in Rubicon-knockdown neurons (**Response Fig. 20** and **Supplemental Figure S6 C-E** of the revised manuscript).

Response Figure 20: (A) Representative western blot from lysates of WT cortical neurons nucleofected with control or *Rubicon* shRNA plasmid, and probed for SCARB2 and LAMP1. (B) Fold change in LAMP1 levels in neurons nucleofected with control or *Rubicon* shRNA plasmid. (C) Fold change in SCARB2 levels in neurons nucleofected with control or *Rubicon* shRNA plasmid. (N= 4 experiments, *p<0.05, Mann-Whitney test, error bars indicate S.E.M.)

To test if there is an effect of lysosomal function upon *Rubicon* knockdown, we examined lysosomal acidification using the pH sensitive dye pHLysRed in tandem with LysoPrime Green, which labels all lysosomes. In this assay, we see a small but not statistically significant increase in the number of total lysosomes per soma, so based on both western blotting and cell staining it is possible that *Rubicon* influences lysosomal biogenesis. However, in a cell washout experiment with Bafilomycin A (see Methods for details), we found that loss of *Rubicon* results in a significant increase in the number of acidic lysosomes/soma marked by pHLysRed. Plotting the ratio of acidic to total lysosomes, we see a significant increase upon *Rubicon* knockdown indicating that *Rubicon* most clearly regulates lysosomal pH and thereby function (**Response Fig. 21** and **Figure 6 E-I** of the revised manuscript).

Response Figure 21: (A) Representative max projections of neurons nucleofected with Ctrl or *Rubicon* shRNA and assayed for lysosomal pH using the protocol represented in Fig:6E. Outline of the neuronal soma for quantification is indicated using dashed lines in cyan (B) Total no. of lysosomes labeled by LysoPrime Green in control or *Rubicon* knockdown neurons (C) No. of acidic lysosomes labeled by pHLys Red in control or *Rubicon* knockdown neurons. (D) Ratio of acidic:total lysosomes quantified by colocalizing pHLys Red and LysoPrime Green

punctae in control or Rubicon knockdown neurons (N=3 experiments, two-tailed unpaired t-test, * p<0.05, error bars indicate S.E.M., scale bars = 5µm.)

We were unable to perform the suggested time course experiment with Magic Red as suggested by the referee, since we found the Magic Red signal to be unstable over time and hence the dye has to be continually present along with the neurons while imaged live in imaging medium. Removal of MagicRed from the imaging media after incubation in maintenance media resulted in loss of signal (**Response Fig. 22**). Fortunately, however, the suggested experiment assaying lysosomal acidification with pHlysRed and LysoPrime Green dyes was successful in demonstrating a role for Rubicon in regulating lysosomal activity.

Response Fig. 22. Representative max projections of neurons nucleofected with Ctrl or *Rubicon* shRNA treated with Magic Red dye and imaged live. *Left*, Magic Red kept in the imaging media after incubation in neuronal maintenance media. *Right*, Magic Red taken out and the neurons washed after incubation prior to imaging live in imaging media. Outline of the neuronal soma is indicated in yellow.

Reviewer #3 (Remarks to the Author):

Comes across as disingenuous in parts with respect to claiming novelty.

A mitochondrial stress response is clearly not a novel concept, see for example mitoCPR work from Angelika Amon and subsequently many others. To this reader the novelty may lie in the observation that this appears to be a neuron specific response. Although they are providing a mitochondrial insult and measuring mitophagy, at no point do they discriminate this from effects on “general autophagy”. The functional characterisation of the proteins is largely an extension of published data to neurons - there is no neuron specific function indicated - so the novelty from an NCB point of view lies principally with Figure 1, which might be questioned (see below).

As the referee points out, discovery of a mitochondrial stress response is not novel per se, and in fact we carefully differentiate the mitophagy stress response pathway we describe here from previous work on the PINK1/Parkin pathway that induces the ubiquitination of damaged mitochondria in response to mitochondrial stress. ***It is important to note that mitoCPR describes changes in mitochondrial import under stress whereas we discuss changes in non-mitochondrially associated proteins induced by mitochondrial damage, whose depletion by the proteasome leads to a significant increase in the efficiency of mitophagic turnover in neurons.*** While potentially complementary cellular responses, these pathways are quite distinct.

Both mitoCPR (identified by the Amon lab- Weidberg et al., Science, 2018) and mitoTAD (identified by the Becker lab- Mårtensson et al., Nature, 2019) pathways are mechanisms that are activated during impaired protein import into the mitochondria. While both these studies are influential, till date there is **no discovery of equivalent stress response pathways in mammalian cells**. Additionally, to date there is **no equivalent mammalian homologue identified for the protein Cis1**, which is a key molecule of the mitoCPR pathway (Uoselis et al., Mol Cell, 2023).

Nevertheless, we have added literature on mitochondrial stress mechanisms that are more relevant to the findings of our study. These responses include

- a. translational attenuation of mitochondrial proteins by deactivation of the transcription elongation factor eIF2 α by the integrated stress response pathway
- b. activation of mitochondrial proteases and chaperones and ROS detoxification enzymes as a part of the mitochondrial unfolded protein response (UPR^{mt})
- c. Proteasomal degradation of mitochondrial proteins in the cytosol during impairment of mitochondrial protein import- as a part of the UPR activated by mistargeting of proteins (UPR^{am}) response.

While these are interesting and potentially important mechanisms, they have not yet been reported to function in mammalian neurons. Moreover, all these responses are elicited to repair damaged mitochondria. Mitophagy serves as one of the last key steps to eliminate damaged mitochondria that can no longer undergo the state of repair. Hence existence of this key mitophagic stress response, specifically in neurons, is particularly important in this non-dividing cell type- as these mechanisms will prevent the build up of toxic mitochondrial fragments over the lifetime of a neuron.

Thus, we are confident that there is no previous description in the literature of the effects of mitochondrial stress on the selective degradation of these negative regulators of mitophagy (Rubicon and MTMR2/5), and no previous description in the literature of the cellular mechanisms by which Rubicon depletion enhances mitophagy in response to acute mitochondrial stress.

We also disagree with the characterization that our work on Rubicon is largely an extension of published data to neurons. In fact, we recently published a study of the effects of Rubicon on autophagy and mitophagy in HeLa cells (Tudorica et al., JCB, 2024), and the results in this simple cell model are quite distinct from our observations in neurons. Rubicon was found to be dispensable for mitophagy in HeLa cells, but in neurons we find that depletion of Rubicon induces a significant increase in mitophagic flux. ***And there is no published study to date discussing the role of Rubicon in regulating lysosomal biogenesis and function.*** Most studies have demonstrated Rubicon to be a suppressor of initial steps of autophagosome biogenesis. But we find that Rubicon is primarily associated with lysosomes in neurons, and determined key functions of this protein in regulating lysosomal physiology and blocking autophagosome maturation.

This paper from Muqit/Harper labs takes a global approach to quantifying the response to Antimycin and oligomycin in mouse cortical neurons, doi: 10.1126/sciadv.abj0722. It is cited in a fleeting way as reference 29. In a 5 hour time period they find few proteins whose levels change, although interestingly they see another myotubularin family member MTM1 going up. The three proteins that apparently change within two hours here (Figure 1A) are all unchanged in that dataset - should this glaring inconsistency with published data not be addressed head on. This other paper has the advantage of taking an unbiased global proteomic view, which is fairly routine these days. In the current paper they go straight to a select group of proteins and find remarkable changes. Some selected proteins are presented as controls, but these do not even include other MTM family members.

Indeed, the paper from Antico et al. (2021) does note that they find that levels of only a few proteins are significantly decreased when they compare lysates of cortical neurons treated with 10 μ M Antimycin A and 1 μ M oligomycin for 5 hours. Importantly, however, their assays of whole cell lysates lacked the resolution to see significant degradation of Mitofusin 2 (Mfn2) under these conditions, which as noted above is a well-characterized early marker of the PINK1/Parkin response to mitochondrial damage (Chen and Dorn, Science, 2013; McLelland et al, 2018, eLife; Gegg et al, Hum Mol Genet 2010). Our study convincingly recapitulates this critical finding from multiple previous studies where we see MFN-2 degradation within 2 hrs upon application of Ant A thereby indicating the initiation of mitochondrial damage under these conditions.

Thus it is not surprising that the study by Antico et al. lacked the sensitivity to see the degradation of Rubicon, MTMR2 or MTMR5 in their whole cell lysate experiments. Further, as these negative

regulators are not mitochondrially-associated proteins, they would not be expected to appear in the higher sensitivity data sets of mitochondrially-enriched proteins, where Antico et al. did see evidence of Mfn2 depletion. So while we agree that unbiased proteomics can be a powerful technique, this is not the only approach that can yield new information on cellular mechanisms.

However, to extend our study, we are happy to more broadly survey the components of the mitophagy and autophagy pathway. Specifically, we examined the effects of acute mitochondrial damage on other proteins in the MTM family as suggested by this referee – please see Fig. 3 in the revised manuscript.

Myotubularins, including MTMR2 are both PI3P and PI3,5P2 phosphatases - this is never indicated - nor is the role of MTMR2 activity in mitophagy ever tested. What is the critical effector.

The role of the MTMR2/MTMR5 complex in neuronal autophagy has been beautifully described in a recent paper by Chua et al. (2022), as we note in our manuscript. We independently confirmed these findings as shown in Figure 6D,E by demonstrating that depletion of MTMR2 has a modest effect on increasing mitochondrial turnover by autophagy. We further highlight the role of the myotubularins in our model depicted in Figure 7. However, as Rubicon depletion had a more dramatic effect on neuronal mitophagy, and since the role of Rubicon as a negative regulator of mitophagy in neurons is not well-understood, we thought this was a more appropriate focus for our studies.

Alternative pathways for mitophagy other than PINK1/Parkin exist- such as BNIP3/NIX which are ubiquitin independent. This has been highlighted in neurons in this paper for example doi: 10.1016/j.molcel.2021.10.001 and is likely the major pathway for mitophagy, albeit if the PINK1 pathway is the most sensitive to mitochondrial damage. The reader of this paper is left in the dark about this.

We are well aware that there are alternative pathways for mitophagy, as we have previously published on NIX (Simpson et al., 2021) and have a manuscript on compensatory pathways induced by PINK1 knockout posted on bioRxiv (Goldsmith et al., 2023). But it is unclear what the referee is asking here, as there is no evidence that these mitochondrially-targeted pathways would selectively degrade Rubicon or MTMR2/5. Further, the elegant work from Ordureau, Harper and colleagues (Ordureau et al., 2021) cited by the referee examines the role of BNIP3/NIX in neuronal differentiation, rather than in the response to mitochondrial damage in fully differentiated neurons, the focus of our current study. However, to comply with the reviewer's comment, we have looked at BNIP3 and found its level to be unchanged during mitochondrial stress in neurons. Please see **Response figure 23** below, and **Figure 3** and **Supplemental Figure S4** of the revised manuscript.

But to summarize,

- We do see a reduction in the level of mitofusin-2 (**A, B**), indicating the initiation of mitochondrial damage. We probed for proteins involved in mitophagy (Parkin, TBK1, RAB7, BNIP3) and we do not see a significant change in the levels of any of these proteins (**C-F**). We also looked at other mitochondrial proteins such as COX II and mitofilin and found their levels to be unaffected during initiation of mitochondrial damage (**G, H**)
- We looked at proteins involved in autophagy (VPS34, ATG5, ATG7, GABARAP, LC3) and we did not see change (**I-M**) in their levels
- Since Rubicon is a lysosomal protein we looked at other lysosomal proteins (SCARB2, LAMP1, vATPase, Cathepsin B), and we did not see any major changes except for Cathepsin B (N-R). This decrease in Cathepsin B levels can result from an increase in lysosomal degradative activity, and thus represent an indirect effect of the induction of Rubicon degradation (M, N).
- We also looked at levels of MTM1 as the reviewer suggested, but do not see any major changes in its levels. We looked at the level of MTMR14 (also called jumpy) another well characterized negative regulator of autophagy (Vergne et al., EMBO, 2009), but we do not see any change in their levels (**S, T**).
- We looked at proteins which are not related to autophagy and are required for housekeeping such as actin, tubulin and GAPDH, but we do not see changes in their levels upon mitochondrial damage (**U-W**).

Response Fig. 23. (A) Representative western blot of wild type cortical neurons treated with vehicle (EtOH) or 30 nM Ant A for 2 hrs and probed for MFN-2 (B) Quantification of MFN-2 levels in neurons treated with vehicle (EtOH) or Ant A. (C-H) Quantification of levels of mitophagy proteins (Parkin, TBK1, RAB7) and mitochondrial proteins (COX II, Mitofilin) in neurons treated with wild type cortical neurons treated with vehicle (EtOH) or 30 nM Ant A for 2 hrs. (I-M) Quantification of levels of autophagy associated proteins (VPS34, ATG, ATG7, GABARAP, LC3) in neurons treated with wild type cortical neurons treated with EtOH or 30 nM Ant A for 2 hrs. (N) Representative western blot of wild type cortical neurons treated with EtOH or 30 nM Ant A for 2 hrs and probed for Cathepsin B (O) Quantification of Cathepsin B levels in neurons treated with EtOH or Ant A. (P-R) Quantification of levels of other lysosomal proteins (SCARB2, LAMP1, ATP6V1E1)

in neurons treated with wild type cortical neurons treated with vehicle (EtOH) or 30 nM Ant A for 2 hrs. **(S, T)** Quantification of levels of other MTM1 and MTMR14 in neurons treated with wild type cortical neurons treated with EtOH or 30 nM Ant A for 2 hrs. **(U-W)** Quantification of levels of house-keeping proteins (Tubulin, GAPDH, Actin) in neurons treated with wild type cortical neurons treated with EtOH or 30 nM Ant A for 2 hrs. (N=3 experiments, $p < 0.05$, error bars indicate S.E.M., scale bars = 5 μ m.) Two-tailed unpaired t-tests were performed to determine significance for most proteins, except for COX II, BNIP3, TBK-I. Mann-Whitney test was performed for TBK-1, COX II and BNIP3 to test for significance as the data were not found to be normal.

These observations indicate that the degradation of MTMR5, MTMR2 and Rubicon via MitoSR represent specific responses to mitochondrial stress in neurons, and not a generalized loss of proteins owing to mitochondrial damage.

Summary of Responses:

In sum, we found the comments of the three referees to be very helpful in further strengthening our data or clarifying the impact of our work. In particular, the addition of further data establishing the selective degradation of negative regulators of mitophagy, as well as the effects of Rubicon depletion on lysosomal activity and autophagosome-lysosome fusion in neurons add to the impact of our work. We think that the revised manuscript represents an advance in our understanding of mitophagy in neurons. In fact, these findings have already led to translational studies by our collaborators to identify Rubicon inhibitors as possible therapeutic treatments for neurodegenerative disease.

Based on the strength of our findings, our extensive responses to the points raised by the three referees, and the resulting significant improvements to our revised manuscript, we hope that this work will be found acceptable for publication in *Nature Communications*.